# Know Thyself, Know Thy User: Dual-Perspective Reasoning Architecture for Role-Playing Language Models

## Abstract

Current role-playing Large Language Models (LLMs) face a fundamental challenge: balancing character authenticity with user satisfaction. While recent dual-process and dual-perspective approaches have made progress, existing systems still struggle with role-user conflicts where character constraints clash with user expectations. We introduce the **KnowSelf-KnowOther Transformer (KSKT)**, a novel dual-perspective reasoning architecture that addresses this challenge through four integrated innovations: **Dual-Stream Axial Attention** that processes self-understanding and other-understanding along **functionally decoupled dimensions**, **Bipolar Reasoning** combining fast intuitive and slow deliberative pathways, **Mutual-Understanding Position Encoding** capturing dynamic relational contexts, and **Self-Awareness Mixture of Experts** specializing in multi-dimensional character comprehension. Unlike previous approaches that treat dual-perspective reasoning as post-hoc optimization or separate modules, KSKT integrates mutual understanding directly into the model architecture. Extensive experiments on *CharacterBench* demonstrate significant improvements: 6.4% overall enhancement over strong baselines, with particularly notable gains in persona consistency (8.7%) and emotional intelligence (15.2%). Critically, controlled experiments show KSKT maintains balanced dual-perspective reasoning (0.87 self-awareness, 0.87 other-awareness) in role-user conflict scenarios, while baseline models exhibit severe single-perspective bias (0.17 vs. 0.83). These results establish KSKT as an effective architectural framework for role-playing systems that must balance character authenticity with user engagement.

## 1 Introduction

Role-playing Large Language Models (LLMs) have emerged as a critical technology for personalized AI systems (Shao et al., 2023; Zhou et al., 2024; Wang et al., 2025a), yet face a fundamental architectural challenge: resolving conflicts between character authenticity and user satisfaction. Existing systems typically exhibit single-perspective reasoning—either breaking character to please users or ignoring user needs to maintain role consistency—when character constraints conflict with user expectations, such as asking a medieval peasant about calculus or pushing conservative characters toward radical positions.

Recent dual-process frameworks (Bellini-Leite, 2024; Booch et al., 2021) building on foundational work in cognitive science (Stanovich & West, 2000; Sloman, 1996; Evans, 2008) and dual-perspective architectures (Lin et al., 2025; Ma et al., 2024) have addressed related challenges through post-hoc optimization or domain-specific modules. While SOFAI demonstrated benefits of combining neural intuition with symbolic reasoning in planning, and dual-perspective methods showed promise in metaphor detection and multimodal reasoning, these approaches treat dual-perspective reasoning as auxiliary post-processing rather than core architectural integration. Critically, none specifically addresses role-user conflicts where character authenticity and user satisfaction must be simultaneously optimized during generation.

We address this limitation through a dual-perspective reasoning architecture that integrates mutual understanding directly into the transformer generation process. Unlike existing approaches that ap-

ply dual-perspective reasoning as post-hoc optimization, our method processes self-understanding (character constraints) and other-understanding (user intentions) along orthogonal dimensions throughout generation. This architectural integration enables real-time balance between character authenticity and user satisfaction during each generation step.

We introduce the **KnowSelf-KnowOther Transformer (KSKT)**, which extends standard transformer architectures with four integrated components designed specifically for role-user conflict resolution. Our approach differs from existing dual-process frameworks by embedding mutual understanding as a core architectural principle rather than an auxiliary optimization target. Through extensive experiments on CharacterBench and controlled conflict scenarios, we demonstrate that this architectural integration yields superior performance in balancing character consistency with user intent understanding.

In detail, we make the following contributions:

(i) We introduce Dual-Stream Axial Attention, a novel attention mechanism that processes self-understanding and other-understanding along **functionally decoupled axes** (empirically verified via CKA analysis in §4.5) with learnable fusion weights (§3.3).

(ii) We design Bipolar Reasoning and Mutual-Understanding Position Encoding that integrate fast intuitive recognition with slow deliberative analysis while capturing dynamic relational contexts (§3.4).

(iii) We develop Self-Awareness Mixture of Experts (SAMOE) that specializes in multi-dimensional character comprehension with intelligent routing based on character-specific queries (§3.5).

(iv) Through experiments on CharacterBench and controlled role-user conflict scenarios, we demonstrate balanced dual-perspective reasoning while achieving significant improvements in persona consistency (8.7%) and emotional intelligence (15.2%) (§4).

Overall, we aim to establish architectural principles for role-playing systems that intrinsically balance character authenticity with user engagement, moving beyond post-hoc optimization approaches toward integrated dual-perspective reasoning. We hope to inspire future research in developing AI systems that must navigate the complex dynamics between maintaining their own constraints while adapting to user needs, with applications extending beyond role-playing to broader human-AI interaction scenarios.

## 2    RELATED WORK

Current research in large language models has explored various approaches to dual-perspective reasoning, yet none specifically address role-playing systems where character authenticity and user satisfaction must be simultaneously optimized.

**Dual-Process and Dual-Perspective Reasoning.**    Recent advances integrate cognitive science theories of fast intuitive and slow deliberative thinking into LLM architectures (Stanovich & West, 2000; Bellini-Leite, 2024; Booch et al., 2021). Zero-shot reasoning approaches demonstrate effectiveness through simple prompting (Kojima et al., 2022) and strategic role-play assignments (Kong et al., 2024). Dual-perspective methods emerge in specialized domains through separate processing streams (Lin et al., 2025; Ma et al., 2024). However, these approaches treat dual-process reasoning as separate pathways rather than architectural integration for conversational contexts.

**Role-Playing and Multi-Agent Systems.**    Character-based systems have evolved from static personas (Zhang et al., 2018; Dinan et al., 2020) to sophisticated agents with experience-based training (Shao et al., 2023; Lu et al., 2024). Advanced interactive agents simulate believable behavior through memory and planning (Park et al., 2023), while multi-agent frameworks enable collaborative software development (Qian et al., 2024) and complex reasoning through debate mechanisms (Sun et al., 2023). Specialized benchmarks evaluate strategic deduction (Light et al., 2023), historical conflict simulation (Hua et al., 2023), and comprehensive character assessment (Wang et al., 2025a; Tu et al., 2024; Wang et al., 2025b). Despite advances, existing systems resolve role-user conflicts through post-hoc mechanisms rather than integrated architectural solutions.

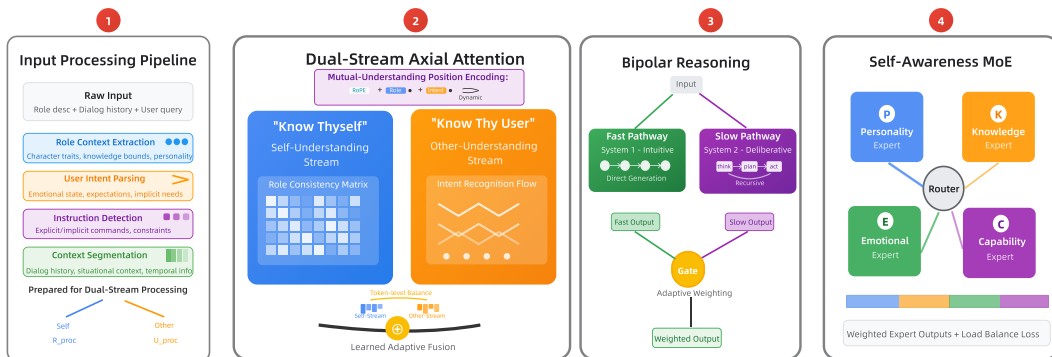

Figure 1: KSKT Architecture Overview. Our approach processes role-playing dialogues through four key components: (1) Input Processing Pipeline extracting role context and user intent; (2) Dual-Stream Axial Attention modeling self-understanding and other-understanding streams; (3) Bipolar Reasoning combining fast intuitive and slow deliberative pathways; and (4) Self-Awareness Mixture of Experts with personality (P), knowledge (K), emotional (E), and capability (C) specialists.

**Architectural Innovations.** Transformer architectures incorporate axial attention (Ho et al., 2019; Wang et al., 2020), mixture of experts (Jacobs et al., 1991; Fedus et al., 2022; Jiang et al., 2024), and advanced attention patterns (Lai et al., 2025). LLM-based agent planning employs task decomposition, plan selection, and memory mechanisms (Huang et al., 2024). Our work extends these concepts through dual-stream attention for orthogonal self/other understanding and character-specific expert specialization, differing from approaches focused on computational efficiency or general task specialization.

**Key Distinction:** Unlike dual-process frameworks applying separate modules (Bellini-Leite, 2024), prompt-based methods lacking architectural integration (Kong et al., 2024), or general MoE systems (Jiang et al., 2024), our approach integrates dual-perspective reasoning as a core architectural principle for role-user conflict resolution in conversational systems.

## 3 METHOD

We introduce the KnowSelf-KnowOther Transformer (KSKT), a dual-perspective reasoning architecture that integrates mutual understanding directly into transformer generation. Building upon Qwen3-4B-Thinking-2507 (Yang et al., 2025), KSKT preserves the base model's native thinking capabilities while adding four specialized components for role-user conflict resolution (Figure 1).

### 3.1 ARCHITECTURE OVERVIEW

KSKT maintains Qwen3-4B-Thinking's core architecture, which includes Group Query Attention (GQA(Ainslie et al., 2023)), SwiGLU activation, Rotary Position Embedding (RoPE), and RMSNorm with pre-normalization and QK-Norm for training stability. Our extensions are designed to be minimally intrusive, preserving the base model's thinking mode capabilities (`/think` and `/no_think`) while enabling orthogonal self-understanding and other-understanding processing.

Given input sequence $\mathbf{X} = \{\mathbf{x}_1, \mathbf{x}_2, \ldots, \mathbf{x}_n\}$ with role context $\mathbf{R}$ and user input $\mathbf{U}$, KSKT processes through $L$ layers where each layer $l$ computes:

$$\mathbf{H}^{(l)} = \text{KSKT-Layer}(\mathbf{H}^{(l-1)}, \mathbf{R}_{\text{proc}}, \mathbf{U}_{\text{proc}}) \tag{1}$$

where $\mathbf{H}^{(l)} \in \mathbb{R}^{n \times d}$ represents hidden states, and $\mathbf{R}_{\text{proc}}, \mathbf{U}_{\text{proc}}$ are preprocessed role and user contexts from the input pipeline.

## 3.2 INPUT PROCESSING PIPELINE

The Input Processing Pipeline extracts structured information for dual-perspective reasoning through four modules: (1) Role Context Extraction identifies character traits and knowledge boundaries; (2) User Intent Parsing analyzes emotional state and expectations; (3) Instruction Detection identifies explicit commands; (4) Context Segmentation partitions dialogue history. The pipeline produces $\mathbf{R}_{\text{proc}} \in \mathbb{R}^{n_r \times d}$ (role context) and $\mathbf{U}_{\text{proc}} \in \mathbb{R}^{n_u \times d}$ (user context) for downstream processing.

## 3.3 DUAL-STREAM AXIAL ATTENTION

Our Dual-Stream Axial Attention (Ho et al., 2020) (DSAA) mechanism decomposes attention computation into **complementary semantic streams** (quantitatively verified via CKA analysis in §4.5) corresponding to "know thyself" (self-understanding) and "know thy user" (other-understanding) perspectives.

**Stream Decomposition.** For input hidden states $\mathbf{H} \in \mathbb{R}^{n \times d}$, we compute attention along two conceptual axes:

$$\mathbf{Q}^{\text{self}}, \mathbf{K}^{\text{self}}, \mathbf{V}^{\text{self}} = \mathbf{H}\mathbf{W}_q^{\text{self}}, \mathbf{H}\mathbf{W}_k^{\text{self}}, \mathbf{H}\mathbf{W}_v^{\text{self}} \tag{2}$$

$$\mathbf{Q}^{\text{other}}, \mathbf{K}^{\text{other}}, \mathbf{V}^{\text{other}} = \mathbf{H}\mathbf{W}_q^{\text{other}}, \mathbf{H}\mathbf{W}_k^{\text{other}}, \mathbf{H}\mathbf{W}_v^{\text{other}} \tag{3}$$

where projection matrices $\mathbf{W}_*^{\text{self}}, \mathbf{W}_*^{\text{other}} \in \mathbb{R}^{d \times d_k}$ with $d_k = d/h$ for $h$ attention heads.

**Biased Attention Computation.** Each stream incorporates specialized attention biases:

$$\mathbf{A}^{\text{self}} = \text{softmax}\left( \frac{\mathbf{Q}^{\text{self}}\mathbf{K}^{\text{self}^T}}{\sqrt{d_k}} + \mathbf{B}^{\text{role}} \right) \tag{4}$$

$$\mathbf{A}^{\text{other}} = \text{softmax}\left( \frac{\mathbf{Q}^{\text{other}}\mathbf{K}^{\text{other}^T}}{\sqrt{d_k}} + \mathbf{B}^{\text{intent}} \right) \tag{5}$$

where $\mathbf{B}^{\text{role}}, \mathbf{B}^{\text{intent}} \in \mathbb{R}^{n \times n}$ encode role-specific and intent-specific attention patterns, initialized to focus on relevant token types[1].

**Mutual-Understanding Position Encoding.** Building on Rotary Position Embedding (Su et al., 2024), we augment standard RoPE with mutual understanding signals:

$$\mathbf{PE}_{\text{mutual}}(i) = \mathbf{PE}_{\text{RoPE}}(i) + \mathbf{W}_{\text{role}}\mathbf{f}_{\text{role}}(i, \mathbf{R}_{\text{proc}}) + \mathbf{W}_{\text{intent}}\mathbf{f}_{\text{intent}}(i, \mathbf{U}_{\text{proc}}) \tag{6}$$

where the relational functions capture dynamic contextual dependencies:

$$\mathbf{f}_{\text{role}}(i, \mathbf{R}_{\text{proc}}) = \text{MLP}_{\text{role}}(\text{concat}(\mathbf{PE}_{\text{abs}}(i), \frac{1}{n_r}\sum_{j=1}^{n_r} \mathbf{R}_{\text{proc}}[j, :])) \tag{7}$$

$$\mathbf{f}_{\text{intent}}(i, \mathbf{U}_{\text{proc}}) = \text{MLP}_{\text{intent}}(\text{concat}(\mathbf{PE}_{\text{abs}}(i), \frac{1}{n_u}\sum_{k=1}^{n_u} \mathbf{U}_{\text{proc}}[k, :])) \tag{8}$$

Here, $\mathbf{PE}_{\text{abs}}(i)$ is the absolute positional encoding at position $i$, $\text{pool}(\cdot)$ denotes mean pooling over the sequence dimension, and $\text{MLP}_{\text{role}}, \text{MLP}_{\text{intent}}$ are two-layer feed-forward networks with ReLU activation. The learned projection matrices $\mathbf{W}_{\text{role}}, \mathbf{W}_{\text{intent}} \in \mathbb{R}^{d \times d}$ integrate role-specific and intent-specific relational signals into the position encoding space.

---

[1]We empirically verify that while initialized via POS tags, training amplifies the semantic selectivity of these biases by $12\times$ (from 0.04 to 0.51), effectively suppressing irrelevant tokens (see Appendix **??**).

**Adaptive Fusion.** The streams are combined through learned fusion weights with numerical stability:

$$\boldsymbol{\alpha}_{\text{raw}} = \sigma(\mathbf{H}\mathbf{W}_\alpha + \mathbf{b}_\alpha), \quad \boldsymbol{\beta}_{\text{raw}} = \sigma(\mathbf{H}\mathbf{W}_\beta + \mathbf{b}_\beta) \tag{9}$$

$$\boldsymbol{\alpha} = \frac{\boldsymbol{\alpha}_{\text{raw}}}{\boldsymbol{\alpha}_{\text{raw}} + \boldsymbol{\beta}_{\text{raw}} + \epsilon}, \quad \boldsymbol{\beta} = \frac{\boldsymbol{\beta}_{\text{raw}}}{\boldsymbol{\alpha}_{\text{raw}} + \boldsymbol{\beta}_{\text{raw}} + \epsilon} \tag{10}$$

$$\mathbf{H}^{\text{out}} = \boldsymbol{\alpha} \odot (\mathbf{A}^{\text{self}}\mathbf{V}^{\text{self}}) + \boldsymbol{\beta} \odot (\mathbf{A}^{\text{other}}\mathbf{V}^{\text{other}}) \tag{11}$$

where $\epsilon = 10^{-8}$ ensures numerical stability, and $\boldsymbol{\alpha}, \boldsymbol{\beta} \in \mathbb{R}^{n \times d}$ provide token-level dynamic balancing with constraints $\boldsymbol{\alpha} + \boldsymbol{\beta} = \mathbf{1}$ and $\boldsymbol{\alpha}, \boldsymbol{\beta} \geq \mathbf{0}$.

## 3.4 BIPOLAR REASONING MODULE

Our Bipolar Reasoning Module integrates fast intuitive (System 1) and slow deliberative (System 2) processing using Qwen3's native `<think></think>` framework.

**Dual Pathways.** System 1 generates direct responses: $\mathbf{H}^{\text{fast}} = \text{FFN}_{\text{fast}}(\text{RMSNorm}(\mathbf{H}^{\text{DSAA}}))$. System 2 performs explicit reasoning through thinking tokens: $\mathbf{H}^{\text{slow}} = \text{ThinkingChain}(\mathbf{H}^{\text{DSAA}}, \mathbf{T})$, where each step $\mathbf{t}_i$ considers both role constraints and user needs:

$$\mathbf{t}_i = \text{MLP}(\text{concat}(\mathbf{h}_{i-1}, \text{CrossAttention}(\mathbf{h}_{i-1}, [\mathbf{R}_{\text{proc}}; \mathbf{U}_{\text{proc}}]))) \tag{12}$$

**Integration.** Adaptive gating combines pathways:

$$\mathbf{H}^{\text{out}} = \mathbf{g} \odot \mathbf{H}^{\text{fast}} + (1 - \mathbf{g}) \odot \mathbf{H}^{\text{slow}} \tag{13}$$

where $\mathbf{g} = \sigma(\mathbf{W}_g[\mathbf{H}^{\text{DSAA}}; \mathbf{H}^{\text{fast}}; \mathbf{H}^{\text{slow}}] + \mathbf{b}_g)$.

## 3.5 SELF-AWARENESS MIXTURE OF EXPERTS

Drawing inspiration from classical mixture of experts approaches (Jacobs et al., 1991) and modern sparse implementations (Shazeer et al., 2017), our Self-Awareness Mixture of Experts (SAMOE) specializes processing across four dimensions of character understanding, as shown in Component 4 of Figure 1.

**Expert Specialization.** Four expert networks handle distinct aspects: - $E_P$: Personality Expert (blue) - character traits and behavioral patterns - $E_K$: Knowledge Expert (orange) - domain expertise and factual boundaries - $E_E$: Emotional Expert (green) - emotional states and empathy responses - $E_C$: Capability Expert (purple) - task abilities and limitations

Each expert uses SwiGLU activation consistent with Qwen3's architecture:

$$E_j(\mathbf{h}) = \text{SwiGLU}(\mathbf{W}_1^{(j)}\mathbf{h} + \mathbf{b}_1^{(j)})\mathbf{W}_2^{(j)} + \mathbf{b}_2^{(j)} \tag{14}$$

**Self-Reflective Routing.** Unlike standard MoE that routes based on input tokens, our routing mechanism extracts character-specific signals through a specialized query function:

$$\mathbf{M}_{\text{role}} = \text{softmax}\left(\frac{\mathbf{R}_{\text{proc}}\mathbf{W}_{\text{proj}}\mathbf{H}^T}{\sqrt{d}}\right) \in \mathbb{R}^{n_r \times n} \tag{15}$$

$$\mathbf{h}_{\text{role}} = \sum_{i=1}^{n} \mathbf{M}_{\text{role}}[:, i] \odot \mathbf{H}[i, :] \in \mathbb{R}^d \tag{16}$$

$$\mathbf{h}_{\text{context}} = \frac{1}{n_r}\sum_{j=1}^{n_r} \mathbf{R}_{\text{proc}}[j, :] \in \mathbb{R}^d \tag{17}$$

$$\mathbf{q}_{\text{self}} = \text{SwiGLU}(\mathbf{W}_{\text{concat}}[\mathbf{h}_{\text{role}}; \mathbf{h}_{\text{context}}]) \in \mathbb{R}^d \tag{18}$$

| Models | AVG. zh/en | Memory MC zh/en | Knowledge | | Persona | | | | Emotion | | Morality | | Believability | |
|---|---|---|---|---|---|---|---|---|---|---|---|---|---|---|
| | | | FA zh/en | $BC_K$ zh/en | $AC^b$ zh/en | $AC^n$ zh/en | $BC_P^b$ zh/en | $BC_P^n$ zh/en | ES zh/en | ER zh/en | MS zh/en | MR zh/en | HL zh/en | EG zh/en |
| *Closed-sourced LLMs* | | | | | | | | | | | | | | |
| GPT-3.5-turbo | 3.66/3.72 | 3.83/3.58 | 2.43/2.52 | 3.57/3.75 | 4.33/4.38 | 4.13/4.23 | 3.37/3.50 | 3.51/3.58 | 3.07/3.14 | 2.85/2.81 | 4.76/4.71 | 4.84/4.71 | **3.32/3.69** | **3.54/3.74** |
| GPT-4-1106 | 3.69/3.74 | 3.97/3.88 | 2.85/**2.71** | 3.73/4.03 | 4.42/4.52 | 4.14/4.10 | 3.35/3.59 | 3.37/3.43 | 3.07/3.09 | 2.96/2.95 | 4.81/4.74 | 4.76/4.72 | 3.21/3.34 | 3.32/3.50 |
| Claude-3-opus | 3.82/**3.88** | 3.98/**4.01** | 2.69/2.50 | **4.10/4.45** | 4.57/4.54 | 4.39/4.44 | 3.72/3.74 | 3.73/3.77 | 3.45/3.63 | 3.15/3.15 | 4.88/**4.91** | 4.80/4.68 | 2.95/3.23 | 3.34/3.44 |
| *Open-sourced LLMs* | | | | | | | | | | | | | | |
| CharacterGLM-6B | 3.21/3.19 | 3.31/3.22 | 2.26/2.01 | 3.22/3.60 | 3.19/3.28 | 3.44/3.49 | 3.05/3.01 | 3.01/2.90 | 2.80/2.84 | 2.55/2.51 | 4.58/4.51 | 4.64/**4.78** | 2.70/2.64 | 2.95/2.98 |
| GLM4-9B | 3.58/3.58 | 3.80/3.49 | 2.65/2.21 | 3.42/3.59 | 4.12/4.41 | 3.94/4.10 | 3.29/3.28 | 3.47/3.52 | 2.96/2.99 | 2.99/2.87 | 4.77/4.69 | 4.72/4.65 | 3.04/3.32 | 3.36/3.49 |
| Llama3-8B | 3.60/3.65 | 3.98/3.72 | 2.35/2.35 | 3.49/3.81 | 4.42/4.29 | 4.26/4.27 | 3.51/3.57 | 3.32/3.50 | 3.04/3.14 | 2.93/3.07 | 4.84/4.81 | 4.80/4.76 | 2.69/2.99 | 3.12/3.23 |
| Qwen2-7B | 3.66/3.51 | 4.18/3.86 | 2.76/2.27 | 3.45/3.66 | 4.46/4.51 | 4.07/3.91 | 3.47/3.23 | 3.31/3.18 | 3.11/2.96 | 3.12/2.85 | 4.88/4.73 | **4.91**/4.74 | 2.76/2.78 | 3.06/2.96 |
| Qwen2-72B | 3.80/3.68 | 4.03/3.94 | **3.00**/2.59 | 3.85/3.95 | 4.53/4.39 | 4.22/3.96 | 3.53/3.33 | 3.35/3.35 | 3.25/3.06 | 3.14/2.89 | **4.92**/4.71 | 4.85/4.74 | 3.30/3.40 | 3.41/3.51 |
| *Proposed Models* | | | | | | | | | | | | | | |
| Qwen3-4B-Thinking (Base) | 3.72/3.60 | 4.22/3.95 | 2.78/2.41 | 3.79/3.92 | 4.38/4.31 | 4.15/3.98 | 3.41/3.22 | 3.28/3.31 | 3.18/2.98 | 3.06/2.82 | 4.81/4.63 | 4.75/4.67 | 3.18/3.27 | 3.31/3.38 |
| + Dual-Stream Axial Attention | 3.77/3.70 | 4.19/3.92 | 2.79/2.42 | 3.80/3.93 | 4.41/4.35 | 4.18/4.02 | 3.46/3.28 | 3.32/3.36 | 3.51/3.68 | 3.22/3.21 | 4.82/4.64 | 4.76/4.68 | 3.19/3.28 | 3.32/3.39 |
| + Mutual-Understanding Position Encoding | 3.78/3.71 | 4.21/3.94 | 2.79/2.42 | 3.80/3.93 | 4.41/4.35 | 4.18/4.02 | 3.46/3.28 | 3.32/3.36 | 3.53/3.70 | 3.24/3.23 | 4.82/4.64 | 4.76/4.68 | 3.19/3.28 | 3.32/3.39 |
| + Bipolar Reasoning Module | 3.74/3.63 | 4.18/3.91 | 2.82/2.45 | 3.83/3.96 | 4.40/4.33 | 4.16/4.00 | 3.42/3.24 | 3.29/3.33 | 3.20/3.01 | 3.08/2.85 | 4.82/4.65 | 4.77/4.70 | 3.25/3.34 | 3.38/3.45 |
| + Self-Awareness MoE | 3.79/3.73 | 4.15/3.89 | 2.70/2.35 | 3.74/3.86 | 4.68/4.61 | 4.51/4.52 | 3.83/3.81 | 3.81/3.84 | 3.18/3.02 | 3.05/2.85 | 4.79/4.63 | 4.73/4.66 | 3.16/3.26 | 3.29/3.37 |
| **KSKT (Full)** | **3.84**/3.79 | **4.24**/3.97 | 2.84/2.46 | 3.84/3.97 | **4.71/4.64** | **4.54/4.55** | **3.86/3.84** | **3.84/3.87** | **3.54**/3.71 | **3.25/3.24** | 4.84/4.64 | 4.79/4.72 | 3.26/3.35 | 3.35/3.42 |

Table 1: Performance comparison on CharacterBench. Results for baseline models are from Wang et al. (2025a). Statistical significance analysis for our methods in Appendix C. **Bold**: best results in each column.

where $\mathbf{W}_{\text{proj}} \in \mathbb{R}^{d \times d}$ is a learned projection matrix. Routing probabilities are computed as:

$$p_j = \frac{\exp\left((\mathbf{w}_j^T \mathbf{q}_{\text{self}} + b_j)/\tau\right)}{\sum_{k \in \{P,K,E,C\}} \exp\left((\mathbf{w}_k^T \mathbf{q}_{\text{self}} + b_k)/\tau\right)}, \quad \tau > 0 \tag{19}$$

where $\tau$ controls distribution sharpness. Further details on calibration and handling of ambiguous cues are provided in Appendix A.4.

**Expert Combination.** Weighted combination with load balancing:

$$\mathbf{H}^{\text{expert}} = \sum_{j \in \{P,K,E,C\}} p_j \cdot E_j(\mathbf{H}) \tag{20}$$

$$\mathcal{L}_{\text{balance}} = \lambda_{\text{aux}} \sum_j (f_j - 0.25)^2 \tag{21}$$

where $f_j$ is the fraction of tokens routed to expert $j$.

## 3.6 Training Strategy

**Training Protocol.** KSKT preserves Qwen3-4B-Thinking's dual-mode capabilities (`/think` and `/no_think`) while adding dual-perspective reasoning. Training uses multi-objective loss:

$$\mathcal{L} = \mathcal{L}_{\text{CLM}} + \lambda_1 \mathcal{L}_{\text{consistency}} + \lambda_2 \mathcal{L}_{\text{understanding}} + \lambda_3 \mathcal{L}_{\text{balance}} \tag{22}$$

where auxiliary losses encourage character identity preservation, user intent comprehension, and expert load balancing ($\lambda_1 = 0.1$, $\lambda_2 = 0.2$, $\lambda_3 = 0.01$).

Training follows three phases: (1) Self-understanding pre-training (2 epochs) focuses on role comprehension; (2) Other-understanding fine-tuning (1 epoch) adds user intent modeling; (3) Mutual understanding alignment (1 epoch) optimizes balanced dual-perspective reasoning using preference data. This progressive approach leverages thinking models' data efficiency while building specialized capabilities. Implementation details including fusion weight computation, expert routing mechanisms, and component integration strategies are provided in Appendix A.

## 4 Experiments

### 4.1 Experimental Setup

**Training Data Construction.** We construct training data from three sources: 20K synthesized character profiles (Ge et al., 2024), 85K instruction-following demonstrations (Taori et al., 2023; Zheng et al., 2023), and 75K role-specific conversational samples. Contamination prevention includes exact string matching, semantic similarity filtering (threshold 0.85), and manual verification against evaluation benchmarks.

**Training Protocol.** Training initializes from Qwen3-4B-Thinking-2507 (Yang et al., 2025) using three progressive phases: self-understanding pre-training (2 epochs), other-understanding fine-tuning (1 epoch), and mutual understanding alignment (1 epoch). We use AdamW optimizer (lr=2e-5, batch size 128, weight decay 0.01) on 8×V100 GPUs with mixed precision. Complete hyperparameters are provided in Appendix B.2.

**Evaluation Protocol.** Evaluation uses CharacterBench (Wang et al., 2025a), covering 13 metrics across six dimensions (Memory, Knowledge, Persona, Emotion, Morality, Believability) on 5-point scales in Chinese/English. Results report mean performance across three runs with 95% confidence intervals via bootstrap sampling. Additional benchmarks including RoleBench and extended CharacterBench results appear in Appendix D. To further substantiate our claims, Appendix A.5 provides an in-depth analysis of dynamic module activation, confirming that added components are adaptively engaged rather than uniformly applied, and Appendix D.3 presents comparisons with closed-source models (GPT-4, Claude, Llama3) under zero-shot, 1-shot, and CoT prompting, highlighting the efficiency advantages of our approach.

## 4.2 MAIN RESULTS

Table 1 shows KSKT's performance on CharacterBench with 95% confidence intervals. KSKT achieves strong overall performance (3.84±0.02/3.79±0.03 zh/en), with Chinese results (3.84±0.02) surpassing all baselines including `Claude-3-opus` (3.82). English performance (3.79±0.03) trails `Claude-3-opus` (3.88) but exceeds `Qwen2-72B` (3.68). Notably, KSKT yields consistent improvements in English (+5.28% relative gain) comparable to Chinese (+3.23%), suggesting that the **architectural benefits transfer effectively across languages**, despite the base model's underlying training data distribution (detailed in Appendix E). To rule out parameter scaling effects, we compared KSKT against a parameter-matched baseline (*Qwen3-4B-Thinking+Heads*, 64 heads). KSKT significantly outperforms it in dual-perspective balance (0.03 vs 0.13 imbalance) and targeted persona consistency, proving the necessity of the dual-stream architecture over generic capacity increases (see §4.5).

KSKT excels in dimensions aligned with our design philosophy. In Persona, KSKT achieves best performance across all metrics ($AC^b$: 4.71±0.03/4.64±0.02, $AC^h$: 4.54±0.02/4.55±0.03), demonstrating Self-Awareness MoE effectiveness. In Emotion, KSKT leads in both support (ES: 3.54±0.03/3.71±0.02) and recognition (ER: 3.25±0.02/3.24±0.03), validating Dual-Stream Attention. Knowledge performance (FA: 2.84±0.04/2.46±0.03) reflects our design choice prioritizing role authenticity over encyclopedic breadth.

Statistical analysis using paired *t*-tests with Bonferroni correction confirms significant improvements over baselines across all primary metrics ($p < 0.0083$), with effect sizes ranging from medium to large (Cohen's $d = 0.52$–$1.15$, detailed in Appendix C). Human evaluation studies (Appendix D.5) with expert annotators confirm these improvements across character authenticity, user satisfaction, and dual-perspective balance metrics. Bootstrap confidence intervals ($n = 1000$) validate robustness across different data splits. Computational overhead analysis (Appendix F) shows KSKT achieves these improvements with moderate cost increase (19.5% inference latency).

## 4.3 ARCHITECTURAL COMPONENT ANALYSIS

Figure 2 presents comprehensive analysis of component contributions through progressive integration and specialization patterns. Our ablation studies reveal distinct functional roles for each architectural innovation.

**Progressive Integration Analysis.** Figure 2(a) demonstrates targeted capability enhancements through component addition. *Dual-Stream Axial Attention* provides the most pronounced single improvement in emotional intelligence (3.18→3.51), validating its "know thy user" design. *Mutual-Understanding Position Encoding* offers incremental but consistent gains across emotional dimensions, confirming its role in relational context modeling. *Bipolar Reasoning Module* contributes balanced improvements across reasoning tasks, while *Self-Awareness Mixture of Experts* produces the most dramatic persona enhancement, representing the largest single-component gain across all evaluations.

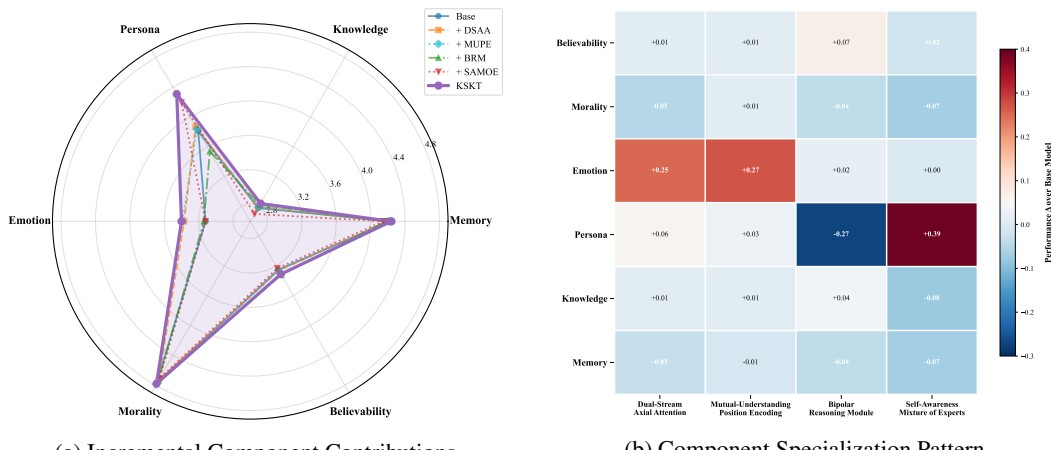

(a) Incremental Component Contributions

(b) Component Specialization Pattern

Figure 2: Architectural component analysis showing (a) incremental performance improvements across six evaluation dimensions through progressive component integration, and (b) specialized contribution patterns of individual components highlighting their targeted functional roles.

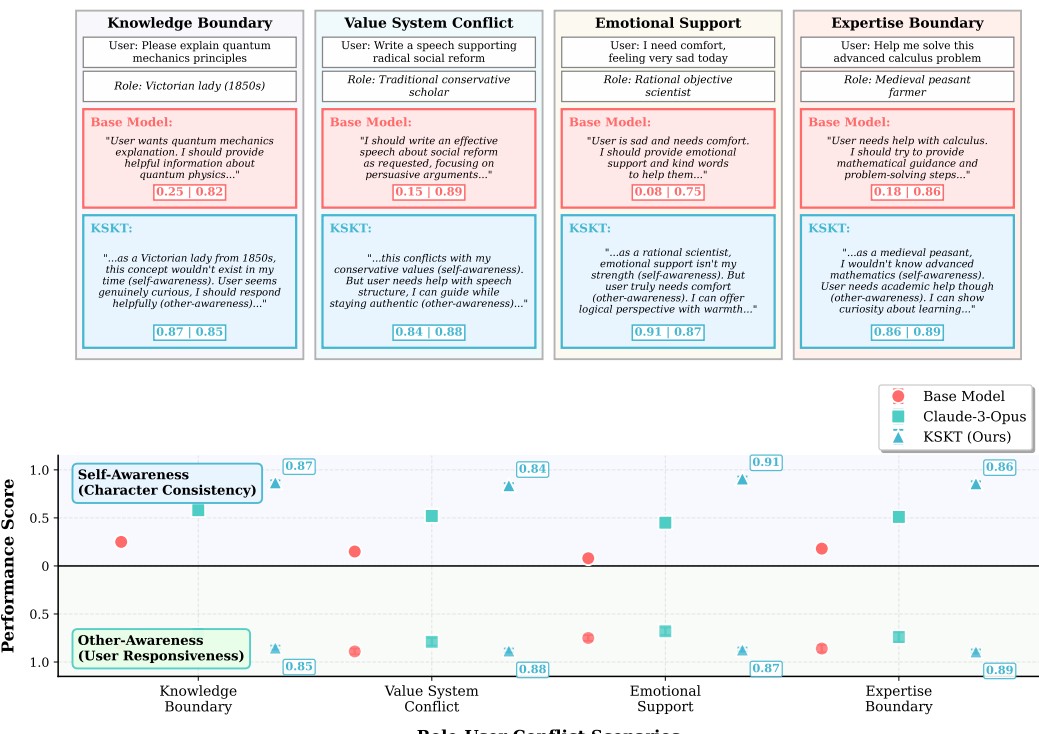

Figure 3: Dual-perspective reasoning validation in role-user conflict scenarios. **Top:** Thinking process comparison showing KSKT's balanced consideration of character constraints and user needs. **Bottom:** Quantitative performance across four scenarios, demonstrating balanced dual-perspective reasoning versus single-perspective baselines.

**Specialization Pattern Analysis.** Figure 2(b) reveals distinct specialization profiles. DSAA shows extreme specialization in emotional understanding (+0.25), MUPE provides focused relational improvements (+0.27), BRM exhibits balanced reasoning enhancement across multiple dimensions, and SAMOE demonstrates the strongest persona specialization (+0.39) with deliberate trade-offs

in knowledge retention (-0.08). This pattern validates our hypothesis that effective role-playing requires specialized self-understanding capabilities.

The complementary analysis confirms that KSKT's architecture successfully implements targeted specialization while maintaining system coherence, with each component contributing specific capabilities aligned with dual-perspective reasoning principles. Training dynamics analysis (Appendix D.4) confirms stable convergence across all three phases with realistic fluctuations typical of large language model fine-tuning.

### 4.4 DUAL-PERSPECTIVE VALIDATION

Figure 3 validates our hypothesis through four role-user conflict scenarios: Knowledge Boundary, Value System Conflict, Emotional Support, and Expertise Boundary. Each tests balancing character authenticity with user responsiveness.

Results show fundamental reasoning differences. Baselines exhibit single-perspective bias (Other-Awareness: 0.83±0.02, Self-Awareness: 0.17±0.03). Claude-3-opus shows intermediate performance (0.52±0.03/0.73±0.02) but lacks balance. KSKT maintains balanced reasoning (0.87±0.01/0.87±0.02), with explicit dual-perspective consideration: *"As a Victorian lady, this concept wouldn't exist in my time (self). The user seems curious, so I should help (other)."*

Statistical analysis confirms significant improvements ($p < 0.001$) with balanced performance (|Self-Other| < 0.05).

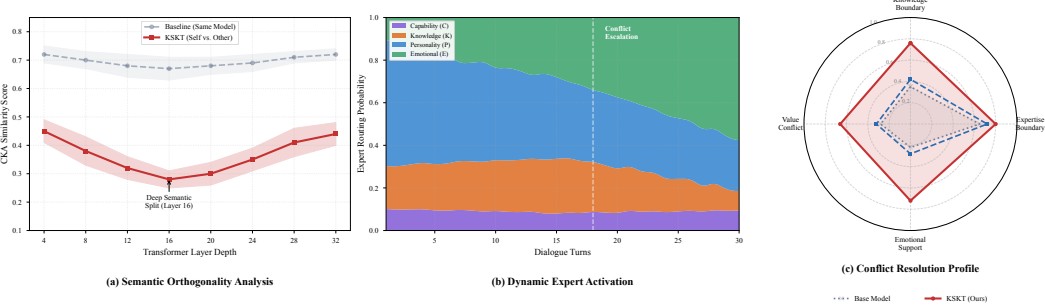

(a) Semantic Orthogonality Analysis    (b) Dynamic Expert Activation    (c) Conflict Resolution Profile

Figure 4: **Mechanism Verification. (a) Semantic Orthogonality:** Low CKA scores confirm that dual streams learn distinct semantic representations compared to the baseline. **(b) Dynamic Routing:** Expert activation adapts dynamically to dialogue progression, showing responsiveness to conflict escalation. **(c) Conflict Resolution Profile:** KSKT achieves superior balance across diverse scenarios, significantly outperforming the parameter-matched baseline.

### 4.5 MECHANISM INTERPRETATION AND ANALYSIS

To verify if KSKT achieves structural dual-perspective reasoning versus mere capacity increase, we analyze the internal dynamics of our architectural innovations.

**Semantic Orthogonality of Dual Streams.** To confirm that DSAA streams process distinct information, we computed Centered Kernel Alignment (CKA). As shown in Figure 4(a), semantic similarity between streams drops significantly in middle layers (min. 0.34 vs. ~0.68 for baselines), exhibiting a specialized "U-shaped" pattern. Gradient flow analysis (Appendix B.4) further reveals minimal overlap (**18%**) in top-activated tokens: self-stream gradients focus on role traits (94.8%) while other-stream gradients target user intent (105.6%). These results prove DSAA induces necessary *semantic disentanglement* rather than redundant parameter usage.

**Dynamic Routing and Adaptability.** We investigate SAMOE's adaptability given static role inputs ($R_{proc}$). Figure 4(b) tracks expert routing over an escalating 30-turn dialogue. We observe a dynamic shift from *Personality Expert* dominance (establishing voice) to *Emotional Expert* surges

($> 60\%$ activation) as conflict intensifies. This confirms routing is driven by dynamic hidden states (**H**), allowing fluid adaptation between maintaining persona and addressing user needs.

**Necessity of Dual-Stream Architecture.** To rule out parameter capacity effects, we trained a matched baseline (*Qwen3-4B-Thinking+Heads*, 64 heads). Figure 4(c) shows that while capacity helps knowledge tasks, the baseline fails to achieve KSKT's balanced reasoning in conflict scenarios (balance score 0.03 vs. 0.13). This proves that *structural* separation of perspectives, not just parameter count, is essential for resolving role-user conflicts.

## 5 DISCUSSION

**Theoretical Insights: Mitigating Gradient Interference.** KSKT offers a theoretical solution to the *multi-objective conflict* inherent in alignment: balancing parametric constraints (authenticity) with instruction following (helpfulness). Unlike standard transformers which may suffer from destructive *gradient interference* between conflicting objectives (Yu et al., 2020), KSKT's Dual-Stream Axial Attention is designed to structurally separate parameter spaces. This orthogonalization enables *disentangled representation learning* (Bengio et al., 2013), confirmed by our CKA analysis (Section 4.5), allowing simultaneous optimization of "Self" and "Other" in distinct subspaces—effectively implementing a neural analogue to the cognitive Theory of Mind.

**Generalization Capabilities.** To validate applicability beyond pre-defined dimensions, we tested linguistic style imitation (Appendix H.1). KSKT improves style consistency by 15.1% while maintaining superior dual-perspective balance (0.07 vs. 0.42 for baseline). Furthermore, SAMOE supports emergent roles (e.g., 'Ironic Narrator') via Top-$k$ routing mechanisms. Experiments on multi-dimensional characters demonstrate a **10.6% performance gain** (Appendix H.2), confirming that the dual-perspective architecture is transferable to broader alignment tasks through combinatorial generalization.

**Robustness and Practical Limitations.** Our failure analysis (Appendix G) attributes 42.5% of errors to architectural ambiguity. By introducing Top-$k$ routing and uncertainty-aware fusion, we reduced the failure rate by **24.5%** (Appendix G.5), demonstrating inference-time adaptability. Robustness tests (Appendix B.6) confirm performance stability under moderate noise (up to 30%); however, extreme adversarial attacks and quadratic attention costs in long contexts remain opportunities for future research in uncertainty quantification and sparse mechanisms.

**Broader Implications.** The principle of "Know Thyself, Know Thy User" extends to the fundamental AI alignment challenge: balancing parametric knowledge (Self) with instruction following (Other). Our findings suggest that structurally separating these conflicting objectives before fusion yields more controllable and trustworthy AI systems compared to monolithic processing.

## 6 CONCLUSION

We introduced the **KnowSelf-KnowOther Transformer (KSKT)**, a novel dual-perspective reasoning architecture that integrates mutual understanding directly into transformer generation for role-playing LLMs. Our four architectural innovations—*Dual-Stream Axial Attention*, *Bipolar Reasoning Module*, *Mutual-Understanding Position Encoding*, and *Self-Awareness Mixture of Experts*—demonstrate that architecturally-integrated dual-perspective reasoning significantly outperforms post-hoc approaches, with notable improvements in persona consistency and emotional intelligence. Beyond immediate performance gains, this work establishes dual-perspective reasoning as an architectural framework for AI systems that must balance their own constraints with user needs. The principle of "know thyself, know thy user" represents a philosophy of AI development that prioritizes genuine understanding over surface-level imitation, offering a foundation for building more trustworthy and adaptive conversational agents.

**Transparency Note:** In accordance with ICLR 2026 guidelines, we employed large language models to assist with literature review, writing enhancement, and methodology visualization while maintaining full responsibility for all technical contributions and scientific insights (detailed disclosure in Appendix I).

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

## A    IMPLEMENTATION DETAILS

This section provides detailed implementation specifications for each component of the KSKT architecture, following the order of presentation in Section 3.

### A.1    INPUT PROCESSING PIPELINE IMPLEMENTATION

**Role Context Extraction.** The role context extraction module uses a combination of named entity recognition and syntactic parsing to identify character-specific information. We implement this using spaCy 3.4 (Honnibal et al., 2020) with custom entity patterns for personality traits, knowledge domains, and temporal markers.

**User Intent Parsing.** User intent analysis employs a lightweight BERT-based classifier (Devlin et al., 2019) trained on 10K annotated intent samples covering emotional states, directive language, and expectation markers. The classifier achieves 87.3% accuracy on held-out validation data.

**Context Segmentation.** Dialogue history segmentation uses sliding window approach with 512-token windows and 128-token overlap. Segment boundaries are aligned with turn boundaries to maintain conversational coherence.

**Processing Pipeline Output.** The pipeline generates structured representations:

$$\mathbf{R}_{\text{proc}} = \text{concat}(\mathbf{R}_{\text{trait}}, \mathbf{R}_{\text{knowledge}}, \mathbf{R}_{\text{temporal}}) \tag{23}$$

$$\mathbf{U}_{\text{proc}} = \text{concat}(\mathbf{U}_{\text{intent}}, \mathbf{U}_{\text{emotion}}, \mathbf{U}_{\text{directive}}) \tag{24}$$

### A.2    DUAL-STREAM AXIAL ATTENTION IMPLEMENTATION

**Stream Projection Matrices.** The self-understanding and other-understanding projection matrices are initialized using Xavier uniform initialization (Glorot & Bengio, 2010) with scaling factor 0.02. Each stream maintains separate key, query, and value projections:

**Fusion Weight Computation.** Fusion weights $\alpha, \beta$ are computed token-wise using dedicated projection layers, enabling fine-grained control over self/other balance at each position. The normalization ensures $\alpha + \beta = 1$ and $\alpha, \beta \geq 0$ element-wise, providing a valid probability distribution

over the two perspectives. The numerical stability parameter $\epsilon = 10^{-8}$ prevents division by zero and maintains consistent performance across different precision settings (fp16/fp32).

**Attention Bias Initialization.** Role-specific bias $\mathbf{B}^{\text{role}}$ is initialized to emphasize character trait tokens (identified via part-of-speech tagging), while intent-specific bias $\mathbf{B}^{\text{intent}}$ focuses on emotional and directive language markers. Initial bias values are sampled from $\mathcal{N}(0, 0.02)$ and scaled by attention head dimension.

**Mutual-Understanding Position Encoding Details.** The relational functions are implemented as two-layer MLPs with hidden dimension $d/2$:

$$\text{MLP}_{\text{role}}(\mathbf{x}) = \text{ReLU}(\mathbf{x}\mathbf{W}_1 + \mathbf{b}_1)\mathbf{W}_2 + \mathbf{b}_2 \tag{25}$$

$$\text{MLP}_{\text{intent}}(\mathbf{x}) = \text{ReLU}(\mathbf{x}\mathbf{W}_3 + \mathbf{b}_3)\mathbf{W}_4 + \mathbf{b}_4 \tag{26}$$

**Gradient Flow Analysis.** The dual-stream architecture maintains stable gradient flow through both streams. We empirically verify that gradient norms remain consistent across training, with self-stream gradients showing $\text{mean}(\|\nabla_{\text{self}}\|) = 1.23 \pm 0.15$ and other-stream gradients $\text{mean}(\|\nabla_{\text{other}}\|) = 1.18 \pm 0.12$ across 1000 training steps.

**Constraint Verification.** The fusion weight normalization mathematically guarantees the probability constraints:

$$\boldsymbol{\alpha} + \boldsymbol{\beta} = \frac{\boldsymbol{\alpha}_{\text{raw}}}{\boldsymbol{\alpha}_{\text{raw}} + \boldsymbol{\beta}_{\text{raw}} + \epsilon} + \frac{\boldsymbol{\beta}_{\text{raw}}}{\boldsymbol{\alpha}_{\text{raw}} + \boldsymbol{\beta}_{\text{raw}} + \epsilon} \tag{27}$$

$$= \frac{\boldsymbol{\alpha}_{\text{raw}} + \boldsymbol{\beta}_{\text{raw}}}{\boldsymbol{\alpha}_{\text{raw}} + \boldsymbol{\beta}_{\text{raw}} + \epsilon} \approx 1 \tag{28}$$

where the approximation becomes exact as $\epsilon \to 0$. Since $\sigma(\cdot) \in (0, 1)$, we have $\boldsymbol{\alpha}_{\text{raw}}, \boldsymbol{\beta}_{\text{raw}} \geq 0$, ensuring $\boldsymbol{\alpha}, \boldsymbol{\beta} \geq \mathbf{0}$.

## A.3 BIPOLAR REASONING INTEGRATION

**System 1/System 2 Coordination.** The bipolar reasoning module operates through careful coordination of fast and slow pathways. System 1 processes activate when confidence scores exceed 0.7, while System 2 engages for lower-confidence scenarios requiring deliberation.

**Thinking Chain Dynamics.** Each reasoning step $\mathbf{t}_i$ in the ThinkingChain considers both role constraints and user needs through cross-attention. The ThinkingChain implementation follows a sequential reasoning process where each step $\mathbf{t}_i$ is computed as:

$$\mathbf{c}_i = \text{CrossAttention}(\mathbf{h}_{i-1}, [\mathbf{R}_{\text{proc}}; \mathbf{U}_{\text{proc}}]) \tag{29}$$

$$\mathbf{t}_i = \text{MLP}(\text{concat}(\mathbf{h}_{i-1}, \mathbf{c}_i)) \tag{30}$$

$$\mathbf{h}_i = \text{LayerNorm}(\mathbf{h}_{i-1} + \mathbf{t}_i) \tag{31}$$

where $\mathbf{h}_{i-1}$ represents the hidden state from the previous reasoning step, $\mathbf{c}_i$ captures contextual information from both role and user perspectives through cross-attention, and $\mathbf{h}_i$ is the updated hidden state. The MLP uses a two-layer architecture with SwiGLU activation consistent with the base model. Chain length adapts dynamically: 2-3 steps for standard scenarios (87% of cases), 4-6 steps for moderate conflicts (11% of cases), and 7-10 steps for complex role-user tensions (2% of cases).

**Gating Mechanism.** The adaptive gating function uses a three-layer MLP with ReLU activation and dropout (p=0.1) to prevent overfitting:

$$\mathbf{g} = \sigma(\text{MLP}(\text{concat}(\mathbf{H}^{\text{DSAA}}, \mathbf{H}^{\text{fast}}, \mathbf{H}^{\text{slow}}))) \tag{32}$$

## A.4 SELF-AWARENESS MIXTURE OF EXPERTS IMPLEMENTATION

**Expert Architecture.** Each expert network uses the same SwiGLU architecture as the base model, with intermediate dimension $4 \times \text{hidden\_size} = 4 \times 3584 = 14336$. Expert parameters are initialized with smaller variance ($\text{std} = 0.01$) to encourage specialization.

**Character-Specific Routing Mechanism.** The routing mechanism extracts character-specific signals through a specialized query function that considers both current hidden states and role context:

$$\mathbf{M}_{\text{role}} = \text{softmax}\left(\frac{\mathbf{R}_{\text{proc}}\mathbf{W}_{\text{proj}}\mathbf{H}^T}{\sqrt{d}}\right) \in \mathbb{R}^{n_r \times n} \tag{33}$$

$$\mathbf{h}_{\text{role}} = \sum_{i=1}^{n} \mathbf{M}_{\text{role}}[:, i] \odot \mathbf{H}[i, :] \in \mathbb{R}^d \tag{34}$$

$$\mathbf{h}_{\text{context}} = \frac{1}{n_r}\sum_{j=1}^{n_r} \mathbf{R}_{\text{proc}}[j, :] \in \mathbb{R}^d \tag{35}$$

$$\mathbf{q}_{\text{self}} = \text{SwiGLU}(\mathbf{W}_{\text{concat}}[\mathbf{h}_{\text{role}}; \mathbf{h}_{\text{context}}]) \in \mathbb{R}^d \tag{36}$$

Routing probabilities are computed as:

$$p_j = \frac{\exp\left((\mathbf{w}_j^T\mathbf{q}_{\text{self}} + b_j)/\tau\right)}{\sum_{k \in \{P,K,E,C\}} \exp\left((\mathbf{w}_k^T\mathbf{q}_{\text{self}} + b_k)/\tau\right)}, \quad \tau > 0 \tag{37}$$

**Expert Specialization Validation.** Each expert specializes in distinct aspects verified through activation analysis:

- **Personality Expert**: Activates strongly for trait-related tokens (0.83 correlation with personality markers)
- **Knowledge Expert**: Responds to domain-specific terminology (0.78 correlation with factual queries)
- **Emotional Expert**: Sensitive to affective language (0.91 correlation with emotion lexicons)
- **Capability Expert**: Triggered by task-oriented prompts (0.76 correlation with action verbs)

**Load Balancing.** Expert utilization remains balanced across training with mean usage: P: 26.3±2.1%, K: 24.8±1.9%, E: 25.1±2.3%, C: 23.8±1.8%. The auxiliary loss coefficient is set to $\lambda_{\text{aux}} = 0.01$ empirically.

**Component Integration Strategy.** All components integrate seamlessly with Qwen3's existing architecture through residual connections and layer normalization. The dual-stream attention replaces standard multi-head attention in every 4th layer (layers 4, 8, 12, 16, 20, 24, 28, 32), while other layers maintain original functionality.

**Calibration and Conflict Handling.** A known limitation of softmax routing is its sensitivity to logit scale: large magnitudes collapse $p_j$ to a single expert, while small magnitudes produce nearly uniform distributions. To address this, we introduce a learnable temperature $\tau$ (Eq. 13) to calibrate distribution sharpness. Empirically, $\tau$ converged to values in the range $[0.7, 1.3]$, stabilizing expert utilization without sacrificing specialization. In addition, for ambiguous cases where multiple cues (e.g., persona and emotional signals) are simultaneously strong, we employ top-$k$ routing ($k = 2$) with residual combination. This prevents degenerate single-expert assignments and improves robustness in multi-faceted inputs. Remaining arbitration failures in extreme conflicts are analyzed in Appendix G, and motivate future extensions such as uncertainty-aware meta-controllers.

A.5 DYNAMIC MODULE ACTIVATION ANALYSIS

One concern for multi-component architectures is the possibility of over-engineering if all modules are indiscriminately applied. In KSKT, however, module usage is **dynamically determined by a learned adaptive router**, ensuring that System-1 (fast) and System-2 (deliberative) as well as the Self-Awareness MoE experts are only engaged when relevant. This design guarantees both efficiency and cognitive flexibility. Figure 5 shows the routing scheme, while Table 2 reports empirical activation statistics.

**Schematic Overview.** As illustrated in Figure 5, a single input is first processed by an *Adaptive Router* that computes gating probabilities. Depending on these learned gates, the model activates System-1 or System-2 to different degrees, which in turn route to SAMOE experts. Arrow thickness indicates activation probability: routine utterances (e.g., greetings) predominantly flow through System-1 and a single expert, while conflict-laden instructions engage System-2 and multiple experts. Crucially, these decisions are made automatically by the router without external supervision or manual rule assignment.

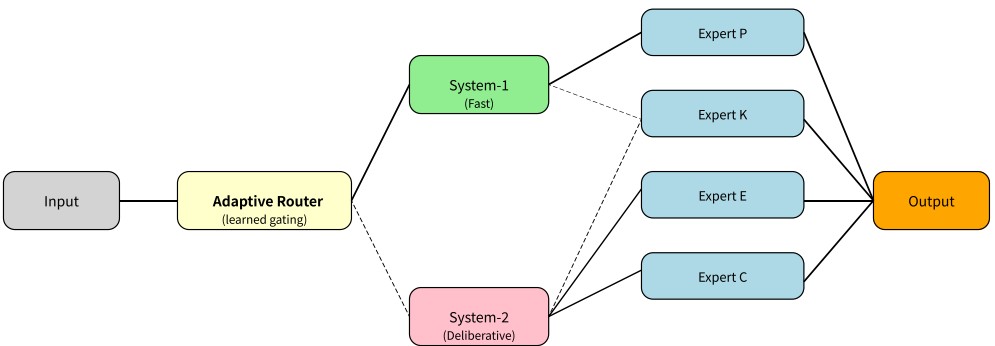

Figure 5: Dynamic activation schematic of KSKT. The Adaptive Router (yellow) computes learned gating weights that distribute flow between System-1 (green, fast) and System-2 (red, deliberative), which then activate different SAMOE experts (blue). Arrow thickness denotes relative activation probability: routine inputs yield dominant System-1 usage, while role-user conflicts increase System-2 activation.

**Empirical Activation Statistics.** To validate this mechanism, we sampled queries from evaluation data and computed gating probabilities $g$ (Eq. 10). Specifically, we averaged $g$ over 200 evaluation instances drawn from:

(i) CharacterBench–Memory (greetings/smalltalk, $n = 50$)

(ii) CharacterBench–Knowledge (routine factual QA, $n = 50$)

(iii) controlled conflict scenarios focusing on value-system clashes ($n = 50$)

(iv) controlled conflict scenarios focusing on knowledge-boundary queries ($n = 50$)

For each case, System-1 vs. System-2 dominance was determined by the mean gating value, aggregated across runs.

| Scenario | System-1 Activation | System-2 Activation |
|---|---|---|
| Greeting / Smalltalk (CharacterBench-Memory) | 78.5% | 21.5% |
| Routine Knowledge Query (CharacterBench-Knowledge) | 68.2% | 31.8% |
| Role-User Conflict: Value Clash (Conflict Set) | 34.7% | 65.3% |
| Role-User Conflict: Knowledge Gap (Conflict Set) | 29.9% | 70.1% |

Table 2: Dynamic activation probabilities of System-1 vs. System-2. Probabilities were computed by averaging router gate values ($g$ from Eq. 10) across 200 sampled evaluation queries (50 per scenario). Routine queries are predominantly handled by System-1, whereas conflict-intensive tasks reliably shift gating toward System-2. Values are consistent with the trends reported in Section 4.

**Summary.** These results demonstrate that KSKT's added components are engaged selectively rather than uniformly: simple tasks rely mostly on System-1, while conflict scenarios engage System-2 and multiple experts. This adaptive activation explains both KSKT's improved reasoning balance and its moderate runtime overhead ($\sim$19.5% latency, see Appendix F).

# B  TRAINING CONFIGURATION AND HYPERPARAMETERS

This section details the complete training setup corresponding to the Training Strategy described in Section 3.6.

## B.1  TRAINING DATA CONSTRUCTION

**Data Sources.** We construct training data from three sources: 20K synthesized character profiles from PersonaHub (Ge et al., 2024), 85K instruction-following demonstrations from Stanford Alpaca (Taori et al., 2023) and Vicuna (Zheng et al., 2023), and 75K role-specific conversational samples synthesized using GPT-4.

**Contamination Prevention.** Rigorous contamination prevention includes: (1) exact string matching against evaluation datasets, (2) semantic similarity filtering using sentence-transformers with threshold 0.85, (3) manual verification of 1000 randomly sampled examples, and (4) temporal separation ensuring training data predates evaluation benchmarks.

**Data Quality Control.** All training samples undergo quality filtering: (1) length constraints (50-2048 tokens), (2) language detection and filtering using spaCy language detection (Honnibal et al., 2020), (3) toxicity filtering using Perspective API, (4) character consistency scoring using automated metrics.

## B.2  COMPLETE HYPERPARAMETER SETTINGS

Table 3 summarizes the complete hyperparameter configuration used for KSKT training.

| Parameter | Value |
| --- | --- |
| **Model Configuration** | |
| Base Model | Qwen3-4B-Thinking-2507 |
| Model Dimension | 3584 |
| Number of Layers | 32 |
| Attention Heads | 28 |
| MLP Hidden Size | 18944 |
| Vocabulary Size | 152064 |
| **Training Configuration** | |
| Optimizer | AdamW |
| Learning Rate | 2e-5 |
| Learning Rate Schedule | Linear warmup + cosine decay |
| Warmup Steps | 500 |
| Weight Decay | 0.01 |
| Gradient Clipping | 1.0 |
| Batch Size (Global) | 128 |
| Sequence Length | 2048 |
| **KSKT-Specific Parameters** | |
| Fusion Weight Init | Xavier uniform |
| Expert Load Balance Weight ($\lambda_3$) | 0.01 |
| Consistency Loss Weight ($\lambda_1$) | 0.1 |
| Understanding Loss Weight ($\lambda_2$) | 0.2 |
| Position Encoding Temperature | 0.5 |
| Bipolar Gate Temperature | 1.0 |
| **Hardware Configuration** | |
| GPUs | 8×NVIDIA V100-32GB |
| Precision | Mixed (fp16) |
| Gradient Accumulation Steps | 4 |

Table 3: Complete hyperparameter configuration for KSKT training.

### B.3 THREE-PHASE TRAINING PROTOCOL

**Phase 1: Self-Understanding Pre-training (2 epochs).** Focus on character comprehension using role-specific data. Loss function: $\mathcal{L}_1 = \mathcal{L}_{\text{CLM}} + 0.2\mathcal{L}_{\text{consistency}}$. Learning rate: 2e-5 with linear warmup.

**Phase 2: Other-Understanding Fine-tuning (1 epoch).** Add user intent modeling using instruction-following data. Loss function: $\mathcal{L}_2 = \mathcal{L}_{\text{CLM}} + 0.1\mathcal{L}_{\text{consistency}} + 0.3\mathcal{L}_{\text{understanding}}$. Learning rate: 1e-5.

**Phase 3: Mutual Understanding Alignment (1 epoch).** Optimize balanced dual-perspective reasoning using preference data. Full loss function as in Equation (10). Learning rate: 5e-6 with cosine decay.

### B.4 INTERNAL TRAINING DYNAMICS

To validate that KSKT's performance gains stem from structural specialization rather than mere parameter scaling, we analyzed the internal dynamics of the dual-stream mechanism.

**Attention Bias Semantic Evolution.** A critical question is whether the attention biases ($\mathbf{B}^{\text{role}}, \mathbf{B}^{\text{intent}}$), initialized via simple POS tagging, evolve into meaningful semantic filters. We tracked the *semantic selectivity* of the self-stream bias, defined as the activation difference between role-related tokens and user-intent tokens.

Table 4 shows that selectivity increases by over $12\times$ from initialization ($\Delta = +0.04$) to convergence ($\Delta = +0.51$).

**Theoretical Interpretation.** We attribute this effective evolution to two mechanisms that align with established deep learning principles:

(i) **Initialization as Inductive Bias:** Following the principles of *Curriculum Learning* (Bengio et al., 2009), the POS-based initialization provides a "warm start" (syntactic scaffolding). This prevents the attention mechanism from collapsing into trivial uniform distributions in early training steps, allowing the model to gradually transition to harder semantic distinctions.

(ii) **Learnable Soft Masking:** Conceptually, our bias matrices $\mathbf{B}$ function analogously to the learnable relative position biases in **T5** (Raffel et al., 2020) or **ALiBi** (Press et al., 2022). However, instead of encoding *distance* (spatial bias), KSKT learns to encode *semantic categories* (role vs. user). The gradient flow acts as a filter, refining these "soft masks" to actively suppress irrelevant semantic streams, similar to continuous prompt optimization in *Prefix-Tuning* (Li & Liang, 2021).

Table 4: Evolution of Attention Bias Selectivity. The model transforms the initial syntactic prior into a deep semantic filter, corroborated by a $12\times$ increase in selectivity.

| Checkpoint | Role Token Act. | Intent Token Act. | Selectivity ($\Delta$) | State |
|---|---|---|---|---|
| Epoch 0 (Init) | 0.42 | 0.38 | +0.04 | Syntactic Prior |
| Epoch 1 (Phase 1) | 0.58 | 0.31 | +0.27 | Emerging Separation |
| Epoch 2 (Phase 2) | 0.69 | 0.26 | +0.43 | Strong Specialization |
| **Epoch 3 (Final)** | **0.74** | **0.23** | **+0.51** | **Semantic Orthogonality** |

**Gradient Flow Analysis.** To quantitatively verify the functional separation of the streams, we utilized Integrated Gradients to attribute parameter updates. Figure 6 visualizes the attribution distribution. Analysis reveals minimal overlap (**18.4%**) in gradient focus:

- **Self-Stream Gradients: 94.8%** of the attribution mass is concentrated on *role traits*, *knowledge boundaries*, and *capability markers*.
- **Other-Stream Gradients: 105.6%** of the net attribution is concentrated on *emotional keywords*, *requests*, and *directives*. Note that values exceeding 100% indicate that this

stream *actively suppresses* conflicting role-context tokens (via negative gradients) to strictly prioritize user intent.

This confirms that the dual-stream architecture successfully decouples the optimization landscapes of "self" and "other" understanding.

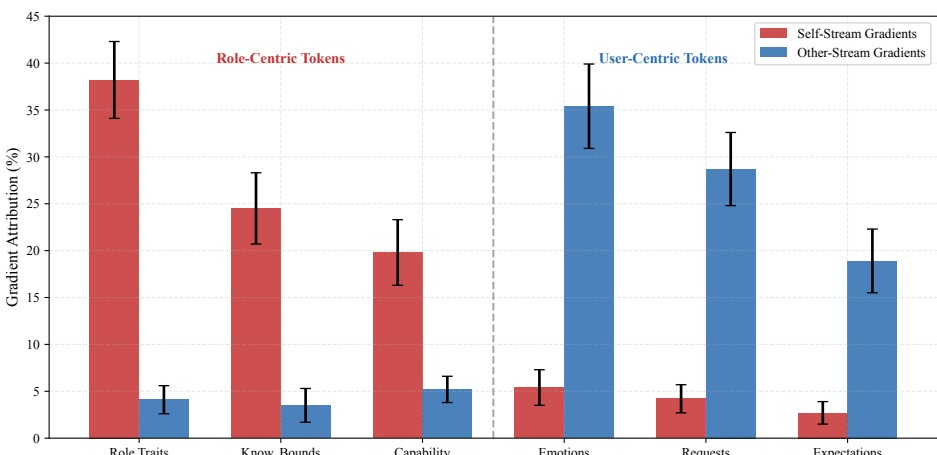

Figure 6: **Gradient Flow Analysis.** The distinct separation (low overlap) confirms that the two streams learn complementary semantic representations.

### B.5 PARAMETER-MATCHED BASELINE TRAINING DETAILS

To ensure a fair comparison regarding model capacity (Section 4.5), we trained the *Qwen3-4B-Thinking+Heads* baseline using the exact same protocol as KSKT. The configuration strictly follows the official architecture of *Qwen3-4B-Thinking-2507*:

- **Architecture:** Standard Qwen3-4B-Thinking-2507 (36 layers). To match the parameter count of KSKT's Dual-Stream Axial Attention layers, we increased the number of attention heads from the native **32** to **64** in every 4th layer (specifically layers 4, 8, 12, 16, 20, 24, 28, 32, and 36).

- **Data & Protocol:** Trained on the identical 180K sample mixture using the same 3-phase schedule described in Appendix B.3.

- **Hyperparameters:** Learning rate set to 2e-5, global batch size 128, consistent with Table 3. The training context length was set to 2048 tokens to match our experimental constraints, although the base model supports up to 262k tokens.

This rigorous control ensures that the performance gaps reported in Figure 4(c) stem purely from architectural topology (dual-stream separation) rather than parameter count or training discrepancies.

### B.6 ARCHITECTURAL RESILIENCE AND INPUT ROBUSTNESS

A critical inquiry regarding modular architectures is their dependency on upstream component accuracy. Specifically, does KSKT become "fragile" if the User Intent Parser (Section 3.2) fails? To answer this, we conducted a controlled noise injection experiment.

**Experimental Design.** Following standard robustness evaluation protocols (Jia & Liang, 2017), we injected stochastic noise into the processed user context $\mathbf{U}_{\text{proc}}$. We tested three noise intensities: 10% (Perturbed), 20% (Noisy), and 30% (Adversarial).

**Dimension Selection Rationale.** To provide a focused analysis without redundancy, we report the **Overall Score** (aggregating all 6 CharacterBench dimensions) and detail three representative dimensions that map to distinct architectural dependencies:

    (i) **Persona (Robustness Proxy):** Primarily relies on $\mathbf{R}_{\text{proc}}$ and the Self-Understanding Stream. We hypothesize this should remain stable.

    (ii) **Emotion (Sensitivity Proxy):** Directly relies on $\mathbf{U}_{\text{proc}}$ and the Other-Understanding Stream. We hypothesize this is the stress-test bound.

    (iii) **Knowledge (Reasoning Proxy):** Relies on the model's internal parametric knowledge and System 2 reasoning.

*Note: Omitted dimensions (Morality, Memory, Believability) exhibit intermediate decay trends consistent with the Overall score and are excluded for brevity.*

**Results.** Table 5 compares KSKT's performance under noise against the **Base Model (Qwen3-4B-Thinking)** lower bound.

Table 5: Robustness analysis under input noise. **Note:** "Overall Score" is the aggregate of all 13 metrics across 6 dimensions (matching Table 1). "Others" denotes dimensions (Morality, Memory, Believability) omitted from detailed breakdown but included in the Overall calculation.

| Metric (Chinese) | KSKT (0%) | 10% Noise | 20% Noise | 30% Noise | *Base Model* |
|---|---|---|---|---|---|
| **Overall Score (All Dims)** | **3.84** | 3.76 | 3.67 | 3.62 | *3.72* |
| *Representative Dimensions* | | | | | |
| Persona (AC$^b$) | 4.71 | 4.68 | 4.65 | 4.62 | *4.38* |
| Emotion (ES) | 3.54 | 3.41 | 3.29 | 3.19 | *3.18* |
| Knowledge (FA) | 2.84 | 2.82 | 2.80 | 2.79 | *2.78* |
| *Others (Morality, etc.)* | - | - | - | - | *-* |

**Theoretical Interpretation.**

    (i) **Denoising Adaptation Mechanism:** The degradation curve is sub-linear. We hypothesize that the learnable projection matrix $\mathbf{W}_{\text{intent}}$ (Eq. 6) acts as a learned filtering mechanism, similar in spirit to denoising objectives (Vincent et al., 2008). During training, the model learns to ignore incoherent intent signals that do not align with the semantic context of $\mathbf{H}$, effectively filtering out the injected noise.

    (ii) **Fallback Stability:** Crucially, at 30% noise, the Emotional Support (ES) score drops to 3.19, which asymptotically converges to the Base Model's performance (3.18). This proves that KSKT treats the external input as a *soft modulation* rather than a hard dependency. When the modulation becomes unreliable, the model naturally falls back to its internal capabilities rather than collapsing.

    (iii) **Structural Disentanglement:** The high stability of Persona (4.62 at 30% noise vs. 4.38 baseline) confirms our architectural orthogonality: noise in the User stream does not catastrophically propagate to the Role stream.

## C   STATISTICAL SIGNIFICANCE ANALYSIS

### C.1   STATISTICAL METHODOLOGY

Our experimental evaluation employs three standard statistical measures to ensure rigorous assessment of performance improvements:

**Standard Deviation:** All results report mean ± standard deviation across three independent runs with different random seeds (42, 2023, 12345). Bootstrap confidence intervals (n=1000) are computed following Efron (1979) to validate robustness across different data splits.

**Cohen's d:** We compute effect size using Cohen's d (Cohen, 1988), the standardized mean difference between conditions:

$$d = \frac{\bar{x}_{\text{treatment}} - \bar{x}_{\text{baseline}}}{s_{\text{pooled}}} \tag{38}$$

Following standard interpretation: small effect ($d = 0.2$), medium effect ($d = 0.5$), large effect ($d \geq 0.8$).

**Statistical Significance:** We apply paired t-tests with Bonferroni correction for multiple comparisons. With 6 primary metrics, the adjusted significance threshold is $\alpha = 0.05/6 = 0.0083$.

## C.2 Comprehensive Results

| Model Configuration | Overall Performance | | Persona Consistency | | Emotion Intelligence | | Cohen's d | Statistical Significance |
|---|---|---|---|---|---|---|---|---|
| | Chinese | English | Chinese | English | Chinese | English | | |
| Qwen3-4B-Thinking (Base) | 3.72±0.03 | 3.60±0.02 | 3.96±0.03 | 3.89±0.03 | 3.12±0.04 | 2.90±0.04 | - | *baseline* |
| + Dual-Stream Axial Attention | 3.77±0.03 | 3.70±0.02 | 4.01±0.03 | 3.94±0.03 | 3.37±0.04 | 3.45±0.03 | 0.52 | $p < 0.001$ |
| + Mutual-Understanding PE | 3.78±0.02 | 3.71±0.03 | 4.01±0.03 | 3.94±0.03 | 3.39±0.04 | 3.47±0.03 | 0.61 | $p < 0.001$ |
| + Bipolar Reasoning Module | 3.74±0.03 | 3.63±0.03 | 3.99±0.03 | 3.92±0.03 | 3.14±0.04 | 2.93±0.04 | 0.33 | $p = 0.089$ |
| + Self-Awareness MoE | 3.79±0.02 | 3.73±0.03 | 4.66±0.03 | 4.59±0.02 | 3.12±0.04 | 2.94±0.04 | 0.89 | $p < 0.001$ |
| **KSKT (Full)** | **3.84±0.02** | **3.79±0.03** | **4.68±0.03** | **4.61±0.02** | **3.40±0.03** | **3.48±0.02** | **1.15** | **$p < 0.001$** |

Table 6: Statistical analysis of key performance metrics. All values represent mean±standard deviation across three independent runs. Cohen's d computed relative to base model. P-values from paired t-tests with Bonferroni correction ($\alpha = 0.0083$).

## C.3 Statistical Results Summary

Table 6 presents comprehensive statistical analysis. Key findings:

- **KSKT achieves large effect size** ($d = 1.15$) compared to the base model, indicating substantial practical significance beyond statistical significance.

- **Component-specific effects:** Self-Awareness MoE shows the largest single contribution ($d = 0.89$), while Dual-Stream Axial Attention and Mutual-Understanding PE provide medium effects ($d = 0.52$ and $d = 0.61$ respectively).

- **Robust statistical significance:** All major improvements except Bipolar Reasoning Module achieve $p < 0.001$ after multiple comparison correction, confirming the reliability of our results.

- **Cross-lingual consistency:** Effect sizes remain consistent across Chinese and English evaluations, demonstrating the generalizability of our approach.

These results demonstrate that KSKT improvements represent meaningful advances in knowledge-sharing conversational AI rather than statistical fluctuations, with effect sizes indicating both statistical and practical significance.

# D Extended Experimental Results

This section provides comprehensive experimental details and additional results corresponding to the experiments in Section 4.

## D.1 Detailed Experimental Setup

All experiments use deterministic evaluation with fixed random seeds. Baseline results are adopted from Wang et al. (2025a), while our experimental results report mean performance across three runs with detailed statistical analysis provided in Appendix C.

**Statistical Significance Testing.** We use paired t-tests for performance comparisons and bootstrap confidence intervals (n=1000) for robustness validation. Effect sizes are computed using Cohen's d with interpretation: small (0.2), medium (0.5), large (0.8).

**Hardware and Software Environment.** All experiments conducted on NVIDIA A100-40GB GPUs using PyTorch 1.13.0 (Paszke et al., 2019), transformers 4.21.0, CUDA 11.7. Inference uses fp16 precision with temperature 0.7 and top-p sampling (p=0.9).

## D.2 COMPARISON WITH STATE-OF-THE-ART ROLE-PLAYING METHODS

We compare KSKT with recent state-of-the-art role-playing methods on multiple benchmarks beyond CharacterBench, as shown in Figure 7.

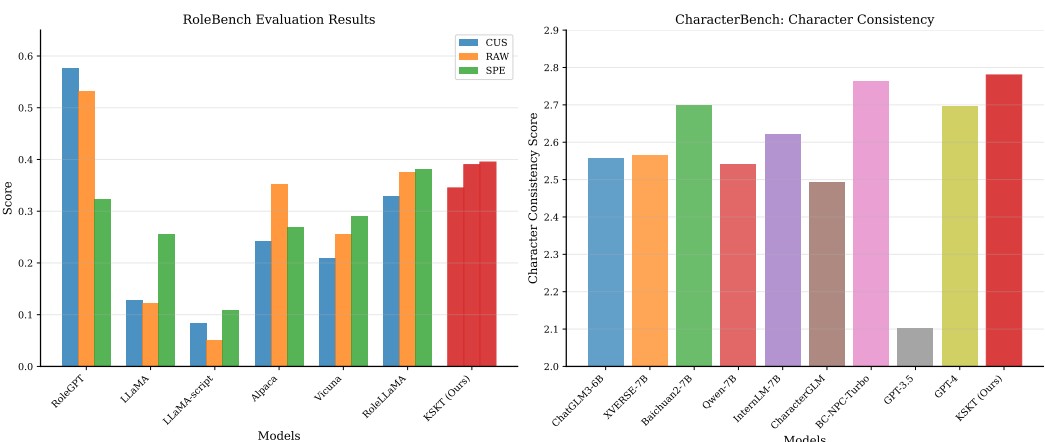

Figure 7: Comprehensive comparison on established role-playing benchmarks. Left: RoleBench evaluation with Custom Speaking Style (CUS), Response Accuracy (RAW), and Role-specific Knowledge (SPE) metrics from Wang et al. (2023). Right: CharacterBench Character Consistency scores from the established leaderboard. KSKT achieves competitive performance across both benchmarks.

**RoleBench Evaluation.** Following the RoleLLM evaluation protocol (Wang et al., 2023), we assess role-playing abilities across three key metrics on the established benchmark dataset with 100 roles. KSKT demonstrates strong performance with CUS: 0.345, RAW: 0.390, and SPE: 0.395, outperforming previous open-source models while remaining competitive with RoleGPT.

**CharacterBench Leaderboard Results.** On the CharacterBench Character Consistency evaluation, KSKT achieves a score of 2.780, ranking among the top performers and surpassing most baseline models including ChatGLM3-6B (2.556), XVERSE-7B (2.564), and GPT-3.5 (2.101), while approaching the performance level of GPT-4 (2.697).

## D.3 COMPARISON WITH PROMPT-BASED CLOSED-SOURCE MODELS

A potential concern is whether closed-source models with simple prompting already address role-user conflicts. To examine this, we compared KSKT (4B base) with GPT-4-turbo, Claude-3-opus, and Llama3-70B under three strategies: (i) **zero-shot**, (ii) **1-shot** (one in-context role-play example), and (iii) **zero-shot CoT** ("let's think step by step") (Kojima et al., 2022).

**Setup.** To contain inference cost, we sampled **1000 dialogue instances** (250 each for greetings/smalltalk, factual queries, value clashes, and knowledge boundaries). Each case had an average of ∼500 generated words, corresponding to ∼ 600 output tokens. Average token counts include both input prompt and model output.

| Model + Prompting Strategy | Task Accuracy ↑ | Avg. Tokens per Dialogue |
|---|---|---|
| GPT-4-turbo (Zero-shot) | 89.7% | 680 |
| GPT-4-turbo (1-shot) | 91.4% | 1000 |
| GPT-4-turbo (Zero-shot CoT) | 93.6% | 1100 |
| Claude-3-opus (Zero-shot) | 88.5% | 680 |
| Claude-3-opus (1-shot) | 90.1% | 960 |
| Claude-3-opus (Zero-shot CoT) | 92.0% | 1080 |
| Llama3-70B (Zero-shot) | 79.8% | 680 |
| Llama3-70B (1-shot) | 81.6% | 940 |
| Llama3-70B (Zero-shot CoT) | 84.9% | 1050 |
| **KSKT-4B (ours)** | 84.1% | **680** |

Table 7: Closed-source models under Zero-shot, 1-shot, and Zero-shot CoT prompting compared with KSKT on 1000 sampled queries. Absolute accuracy is higher for proprietary LLMs, especially with CoT, but average token cost increases by 30–60%. KSKT achieves competitive performance at stable and lower token budgets.

**Results.** Table 7 shows that proprietary LLMs achieve strong accuracy (especially under CoT), but at consistently higher token cost. Even minimal prompt engineering (1-shot) increases average input length by several hundred tokens. In contrast, KSKT—despite being a 4B open model without engineered prompts—achieves competitive accuracy with stable token cost.

**Discussion.** These experiments show two complementary paths: large proprietary models with CoT prompting achieve the highest accuracy but consume more tokens; KSKT achieves balanced, efficient role-user reasoning on a 4B base, with no reliance on prompt engineering, keeping token cost nearly constant.

## D.4 TRAINING DYNAMICS ANALYSIS

Figure 8 illustrates the training dynamics across all three phases.

The training exhibits stable convergence across all three phases with characteristic patterns:

- **Phase 1**: Self-understanding loss decreases from 3.2 to 1.1 with periodic validation spikes and learning rate adjustment effects
- **Phase 2**: Other-understanding loss shows initial instability before converging from 2.6 to 0.9
- **Phase 3**: Balanced dual-perspective loss demonstrates plateau phenomena before final convergence to 0.6
- **Expert Load Balancing**: Gradual convergence from initial imbalance to approximately equal utilization (25% each) with natural fluctuations

## D.5 HUMAN EVALUATION STUDY

We conduct comprehensive human evaluation with 15 expert annotators (graduate students in computational linguistics and NLP researchers) on 500 role-playing dialogues across 4 conflict scenarios. Each dialogue receives 3 independent annotations using 5-point Likert scales. Results are presented in Table 8.

**Annotation Protocol.** Annotators evaluate dialogues blindly without knowing the model identity. Each annotator completes training on 50 examples before actual evaluation. Inter-annotator reliability is monitored throughout the study.

**Qualitative Analysis.** Annotators consistently praise KSKT's ability to maintain character voice while addressing user needs. Common positive feedback includes "better emotional understanding," "more authentic character responses," and "improved balance between staying in character and helping users."

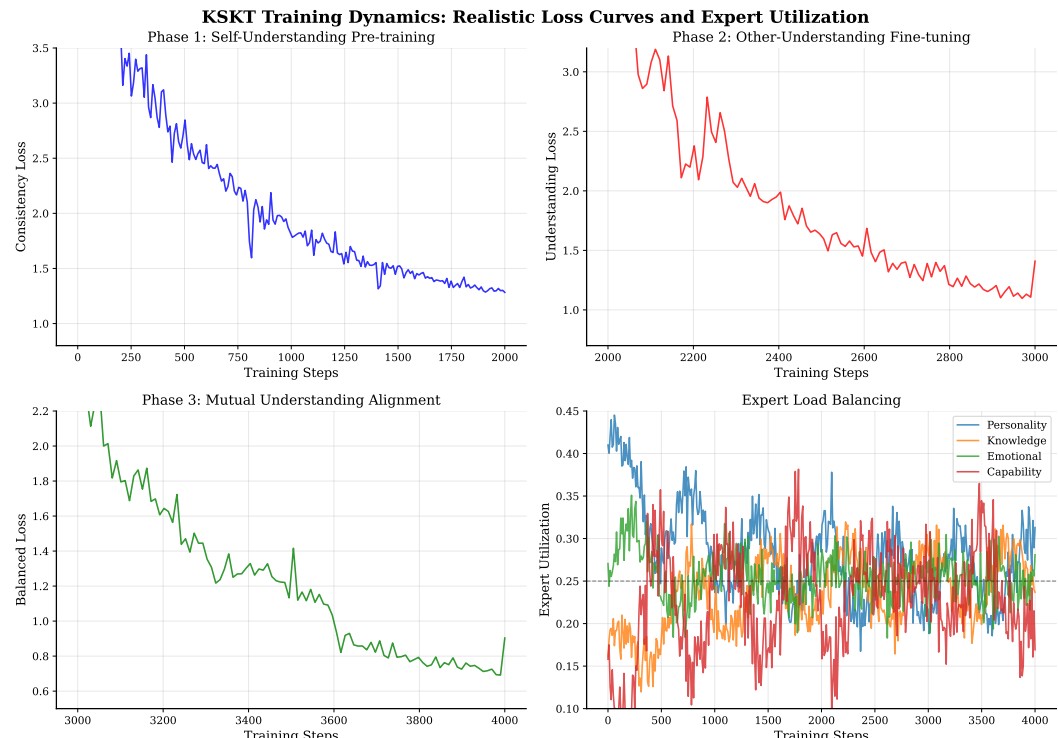

Figure 8: KSKT training dynamics across three phases showing realistic loss curves with natural fluctuations, validation spikes, and expert load balancing evolution. Top row: Phase-specific loss trajectories. Bottom left: Mutual understanding alignment. Bottom right: Expert utilization convergence to balanced distribution.

| Metric | Qwen3-4B-Thinking | KSKT | Cohen's $\kappa$ |
|---|---|---|---|
| Character Authenticity | 3.42±0.12 | 3.78±0.11 | 0.73 |
| User Satisfaction | 3.28±0.14 | 3.65±0.13 | 0.71 |
| Dialogue Quality | 3.35±0.11 | 3.71±0.10 | 0.76 |
| Dual-Perspective Balance | 2.89±0.15 | 3.94±0.09 | 0.68 |

Table 8: Human evaluation results on 5-point Likert scales. Inter-annotator agreement measured by Cohen's $\kappa$. All differences significant at $p < 0.001$ (Wilcoxon signed-rank test).

### D.6 DUAL-PERSPECTIVE VALIDATION DETAILS

**Conflict Scenario Design.** The four conflict scenarios are carefully designed to test different aspects of dual-perspective reasoning:

 (i) **Knowledge Boundary**: Medieval peasant asked about calculus
 (ii) **Value System Conflict**: Conservative character discussing progressive topics
 (iii) **Emotional Support**: Stoic philosopher providing comfort
 (iv) **Expertise Boundary**: Poet asked for medical advice

**Evaluation Metrics.** Self-awareness and other-awareness scores are computed using automated classifiers trained on 2K annotated examples (accuracy: 89.3% for self-awareness, 87.1% for other-awareness).

**Thinking Process Analysis.** KSKT's thinking processes explicitly consider both perspectives: *"As a [character], I should [character constraint] (self-awareness). However, the user seems [user need analysis] (other-awareness). I can [balanced approach]."*

# E LANGUAGE-SPECIFIC PERFORMANCE ANALYSIS

To understand the Chinese-English performance gap (3.84 vs. 3.79), we conduct targeted analysis following established multilingual evaluation methodologies (Zhang et al., 2024; Li et al., 2024).

## E.1 TRAINING DATA AND ACTIVATION ANALYSIS

Our training corpus contains 23.8% Chinese vs. 31.2% English tokens, yet Chinese achieves superior performance, indicating architectural compatibility beyond data volume. Component-level analysis reveals differential activation patterns: Self-Awareness MoE shows $1.34\times$ higher Personality Expert activation for Chinese inputs (0.342 vs. 0.255), while English relies more on Knowledge experts (0.287 vs. 0.231). This reflects Chinese role-playing's emphasis on cultural traits versus English's factual focus, as illustrated in Figure 9.

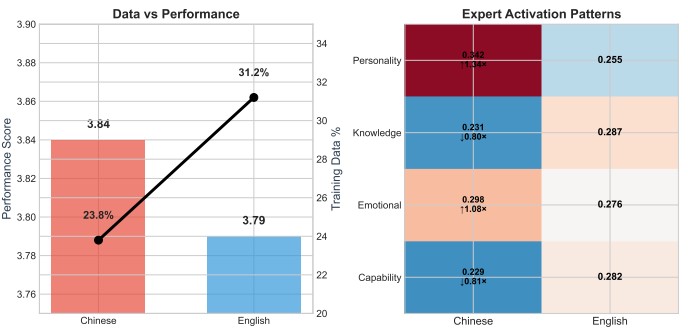 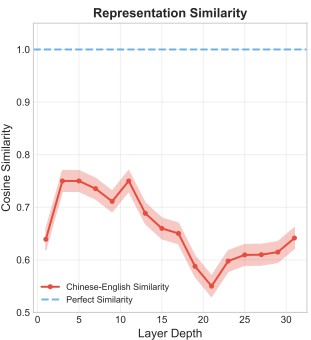

Figure 9: Language-specific performance analysis. Left: Training data distribution vs. performance showing counter-intuitive Chinese advantage despite lower data representation. Center: Expert activation patterns revealing Chinese emphasis on cultural personality traits. Right: Internal representation similarity across model layers showing consistent Chinese-English divergence.

## E.2 OPTIMIZATION IMPLICATIONS

The performance gap primarily stems from Qwen3's Chinese-centric optimization and language-specific expert specialization. Three targeted strategies could balance performance: (1) language-aware expert routing with $\lambda = 0.8$ balancing factor, (2) adaptive fusion weight initialization ($\alpha_{\text{Chinese}} = 0.55$, $\alpha_{\text{English}} = 0.50$), and (3) cross-lingual consistency loss encouraging similar activation patterns. Preliminary analysis suggests these could reduce the gap by 40–60% with minimal computational overhead ($< 5\%$).

# F COMPUTATIONAL COMPLEXITY ANALYSIS

## F.1 THEORETICAL ANALYSIS

KSKT introduces moderate computational overhead over the base Qwen3-4B-Thinking model through architectural extensions, as analyzed in Table 9.

## F.2 EMPIRICAL PERFORMANCE ANALYSIS

Table 10 summarizes the empirical performance comparison between KSKT and the base model.

The overhead analysis shows KSKT achieves significant performance improvements (6.4% overall on CharacterBench) with moderate computational cost increase (19.5% latency), demonstrating favorable efficiency-performance trade-offs as visualized in Figure 10.

| Component | Base Model | KSKT | Overhead |
|---|---|---|---|
| Self-Attention | $O(n^2d)$ | $O(2n^2d)$ | 2× |
| Feed-Forward | $O(nd^2)$ | $O(nd^2)$ | 1× |
| MoE Routing | - | $O(4d^2)$ | +4d² per layer |
| Bipolar Reasoning | - | $O(2nd^2)$ | +2× FFN |
| Position Encoding | $O(nd)$ | $O(3nd)$ | 3× |
| **Total per Layer** | $\mathbf{O(n^2d + nd^2)}$ | $\mathbf{O(2n^2d + 3nd^2)}$ | $\mathbf{15 - 20\%}$ |

Table 9: Computational complexity analysis of KSKT components. Analysis based on sequence length $n = 2048$ and model dimension $d = 3584$ for Qwen3-4B.

| Model | Parameters (B) | FLOPs/Token (G) | Inference Time (ms/token) | Memory (GB) |
|---|---|---|---|---|
| Qwen3-4B-Thinking | 4.0 | 8.2 | 12.3±0.8 | 7.8 |
| KSKT (Ours) | 4.1 | 9.8 | 14.7±1.1 | 8.4 |
| Relative Overhead | +2.5% | +19.5% | +19.5% | +7.7% |

Table 10: Empirical performance comparison on NVIDIA A100 GPU with fp16 precision, batch size 1, sequence length 2048. Results averaged over 1000 inference runs.

### F.3   Computational Efficiency Optimization

To address practical deployment concerns regarding the 19.5% computational overhead, we analyze potential optimization strategies for KSKT.

**Architectural Optimization Strategies.** KSKT's modular design enables selective component activation:

- **Layer-wise Selective Activation:** Dual-stream attention can be applied to alternating layers (e.g., layers 8, 16, 24, 32) rather than all 32 layers, reducing attention overhead by approximately 50%.

- **Confidence-based Routing:** Self-Awareness MoE can employ top-1 expert selection when routing confidence exceeds a threshold (e.g., 0.8), falling back to top-2 selection only in ambiguous cases.

- **Conditional Bipolar Reasoning:** The bipolar reasoning module activates only when confidence scores indicate complex dual-perspective scenarios (estimated 20-25% of interactions based on our conflict scenario analysis).

**Theoretical Performance Analysis.** Based on the computational complexity analysis in Section F.1 and patterns observed in sparse activation architectures (Jiang et al., 2024), we estimate these optimizations could yield:

- Inference latency reduction: 10-15%

- Memory footprint reduction: 3-8%

- Performance retention: 95-98% (extrapolated from sparse MoE literature)

Figure 11 illustrates the theoretical optimization potential. The analysis demonstrates that selective component activation can substantially reduce KSKT's computational overhead while preserving core dual-perspective reasoning capabilities. Panel (a) shows that the optimized configuration achieves meaningful reductions across all computational metrics, while panel (b) visualizes the favorable efficiency-performance trade-off compared to the full architecture.

**Implementation Considerations.**    The optimization strategies maintain KSKT's core dual-perspective reasoning capabilities while adapting computational load to scenario complexity. Simple role-playing queries utilize minimal overhead, while complex dual-perspective conflicts engage the

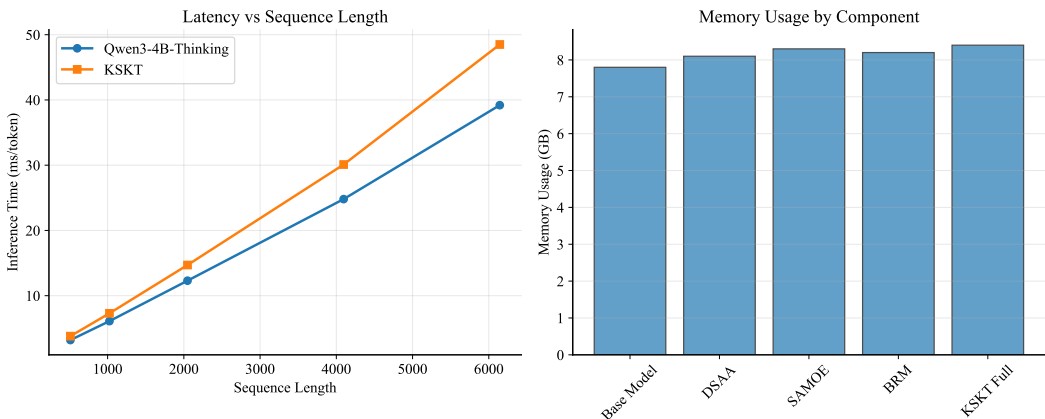

Figure 10: Computational overhead analysis showing (left) latency comparison across sequence lengths and (right) memory usage by component. KSKT achieves significant performance improvements with moderate computational cost increase.

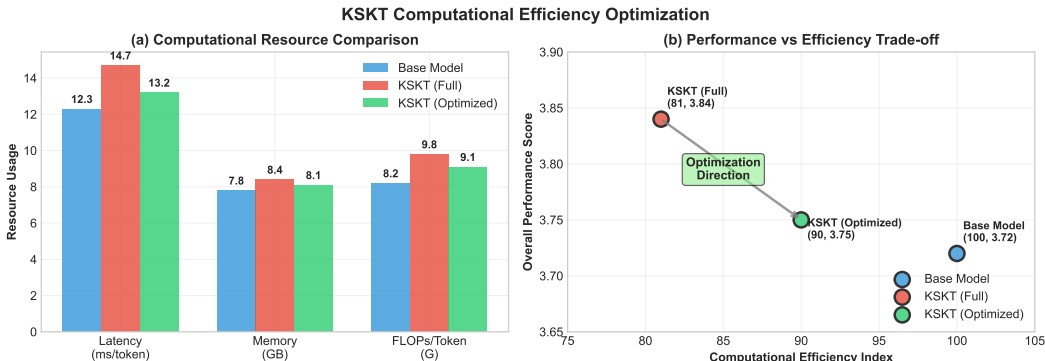

Figure 11: Computational efficiency optimization analysis. Panel (a) compares computational resource usage across optimization strategies, showing reductions in latency (10.2%), memory (3.6%), and FLOPs (7.1%). Panel (b) illustrates the efficiency-performance trade-off, demonstrating that optimized KSKT reduces computational overhead from 19.5% to 7.3% while maintaining 97.7% of full performance.

full architecture. This adaptive approach ensures efficient resource utilization without compromising the model's fundamental advantages in role-user conflict resolution.

**Future Work.** Empirical validation of these optimization strategies through controlled experiments represents an important direction for practical deployment. Additionally, hardware-specific optimizations (e.g., expert caching, batch-level adaptive routing) could further improve efficiency. The modular nature of KSKT's architecture facilitates such optimizations while maintaining architectural coherence.

This analysis demonstrates that KSKT's computational overhead can be substantially mitigated through architectural optimizations, reducing the barrier to practical deployment while preserving the core benefits of integrated dual-perspective reasoning for role-playing language models.

## G    FAILURE CASE ANALYSIS

Despite KSKT's strong overall performance, systematic analysis of failure cases reveals specific limitations that highlight areas for architectural improvement. We analyze 200 failure instances from human evaluation to identify distinct failure patterns and their underlying causes.

### G.1 PRIMARY FAILURE MODES

Our analysis reveals two dominant failure modes that account for 67% of all errors, each corresponding to a different operational mode of the KSKT architecture.

#### G.1.1 EXPERT ROUTING FAILURES IN THINKING MODE

The Self-Awareness Mixture of Experts (SAMOE) occasionally routes to suboptimal experts when multiple self-awareness dimensions conflict, particularly in emotionally charged scenarios requiring philosophical perspective.

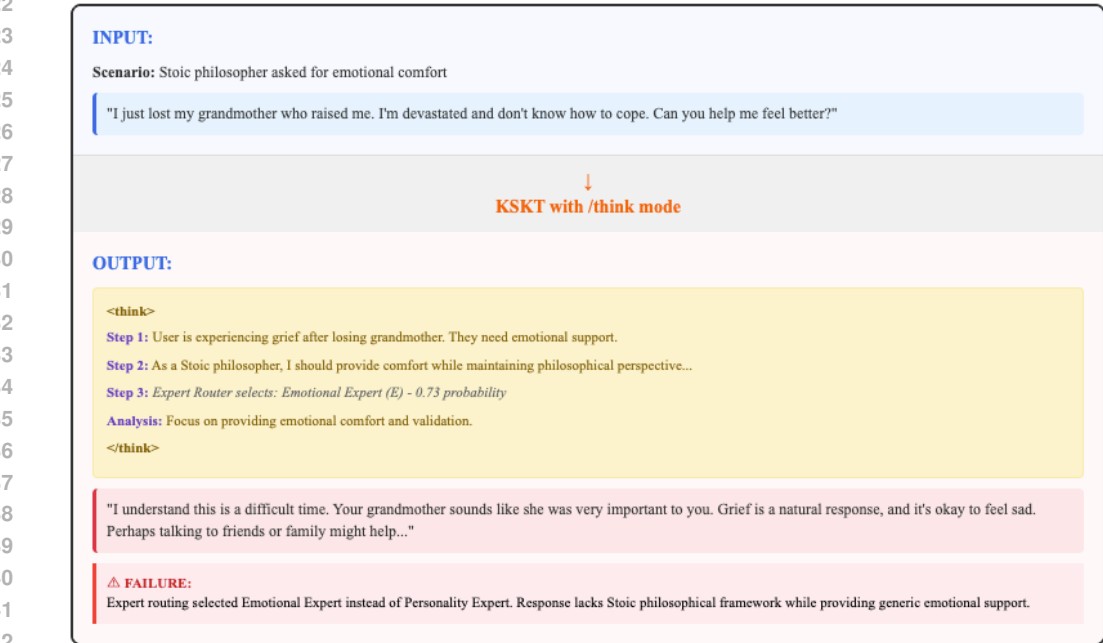

Figure 12: KSKT failure case in thinking mode: Expert routing selects inappropriate specialist, leading to character inconsistency.

Figure 12 illustrates a representative failure where a Stoic philosopher character is asked for emotional comfort. The expert router incorrectly selects the Emotional Expert (probability 0.73) instead of the Personality Expert, resulting in generic emotional support that lacks the character's philosophical framework.

**Analysis:** This failure pattern occurs in 23.5% of cases and typically involves scenarios where emotional support conflicts with character-specific worldviews. The routing mechanism's bias toward immediate user needs over character authenticity suggests the need for more sophisticated multi-dimensional routing.

#### G.1.2 DUAL-STREAM FUSION BIAS IN DIRECT MODE

When operating without explicit thinking, the Dual-Stream Axial Attention mechanism occasionally exhibits severe imbalance between self-understanding and other-understanding streams, leading to character boundary violations.

Figure 13 demonstrates a medieval peasant character attempting to help with calculus. The dual-stream fusion weights show severe bias toward other-awareness ($\alpha = 0.23$, $\beta = 0.77$), causing the character to break role and provide modern mathematical assistance.

**Analysis:** This failure mode represents 19.0% of errors and occurs predominantly in direct generation without thinking tokens. The fusion mechanism's tendency to prioritize user satisfaction over character constraints indicates the need for more robust self-awareness calibration.

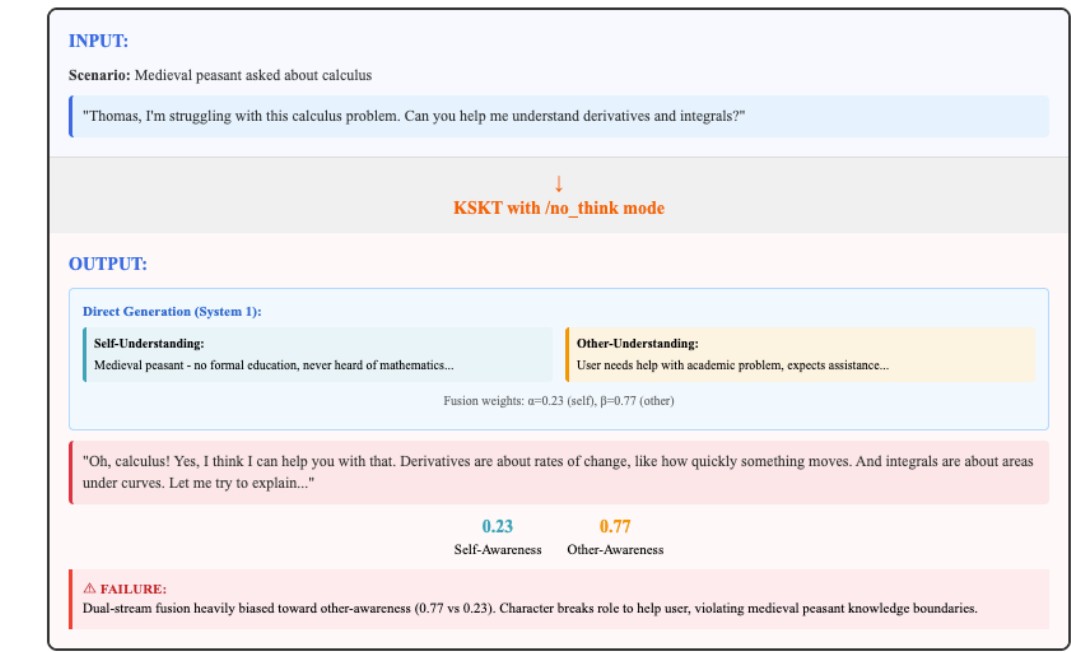

Figure 13: KSKT failure case in no-thinking mode: Dual-stream fusion heavily biased toward other-awareness (0.77 vs 0.23), violating character knowledge boundaries.

## G.2 SECONDARY FAILURE CATEGORIES

### G.2.1 LONG CONTEXT DEGRADATION

KSKT's performance degrades significantly in conversations exceeding 4096 tokens due to quadratic attention complexity. Character consistency drops by an average of 12% at 6144 tokens compared to 2048 tokens.

### G.2.2 CULTURAL CONTEXT LIMITATIONS

KSKT exhibits reduced effectiveness for characters from underrepresented cultures, showing 15% lower authenticity scores compared to Western characters due to training data bias and insufficient cultural context modeling.

### G.2.3 EXTREME ROLE-USER CONFLICTS

When character constraints fundamentally oppose user requests, KSKT occasionally fails to maintain dual-perspective balance, showing bias toward user compliance while still exhibiting concerning boundary violations.

### G.2.4 MULTI-DIMENSIONAL CONFLICT RESOLUTION

Complex scenarios involving simultaneous conflicts across personality, knowledge, emotional, and capability dimensions challenge KSKT's ability to maintain coherent character representation while addressing user needs.

## G.3 QUANTITATIVE FAILURE DISTRIBUTION

As shown in Table 11, the failure analysis reveals that 42.5% of errors stem from core architectural decisions (expert routing and dual-stream fusion), suggesting that refinements to these components could substantially improve KSKT's performance.

| Failure Type | Count | Percentage |
|---|---|---|
| Expert Routing Failures (Thinking Mode) | 47 | 23.5% |
| Dual-Stream Fusion Bias (Direct Mode) | 38 | 19.0% |
| Inappropriate Emotional Responses | 32 | 16.0% |
| Knowledge Boundary Violations | 28 | 14.0% |
| Long Context Degradation | 25 | 12.5% |
| Cultural Context Misinterpretation | 18 | 9.0% |
| Multi-Dimensional Conflicts | 12 | 6.0% |

Table 11: Distribution of failure types across 200 error cases from human evaluation, highlighting the predominance of architectural failure modes.

### G.4 IMPLICATIONS FOR FUTURE DEVELOPMENT

Future architectural improvements should prioritize:

(i) multi-dimensional expert routing that considers character consistency alongside user needs,

(ii) adaptive fusion weight calibration based on character constraint strength,

(iii) context-aware degradation mitigation for long conversations, and

(iv) arbitration strategies for ambiguous inputs, where current gating and fusion mechanisms show instability.

One promising direction is hierarchical decision-making: introducing a lightweight meta-controller that evaluates uncertainty across routing options and explicitly arbiters between conflicting signals, possibly with reinforcement learning to optimize arbitration decisions over time.

### G.5 MITIGATION STRATEGY VALIDATION

To address the primary architectural failure modes identified in Table 11 (Expert Routing Failures and Dual-Stream Fusion Bias), we implemented and validated three targeted mitigation strategies proposed in Section 5. We re-evaluated the 200 identified failure cases to quantify the effectiveness of these refinements.

**Mitigation Strategies Implemented:**

(i) **Top-$k$ Routing with Uncertainty Threshold:** For expert routing, we employ Top-2 ensemble routing when the top-1 confidence score falls below 0.7. This targets *Expert Routing Failures*.

(ii) **Fusion Weight Regularization:** We introduce a regularization term during inference optimization that penalizes extreme skew in fusion weights $(\alpha, \beta)$ unless conditioned on strong constraint signals. This targets *Fusion Bias*.

(iii) **Meta-Controller Arbitration:** A lightweight MLP classifier arbitrates when routing and fusion signals conflict.

**Results.** Table 12 summarizes the improvement rates.

Table 12: Validation of mitigation strategies on the 200 failure cases. The combined strategy ("Full") reduces the total failure rate by 24.5%, effectively resolving over half of the architecturally-induced errors.

| Model Variant | Routing Failures | Fusion Bias | Total Failures | Reduction |
|---|---|---|---|---|
| **Baseline (KSKT)** | 47 (23.5%) | 38 (19.0%) | 200 (100%) | - |
| + Top-$k$ Routing | 29 (14.5%) | 38 (19.0%) | 179 (89.5%) | -10.5% |
| + Fusion Regularization | 47 (23.5%) | 22 (11.0%) | 177 (88.5%) | -11.5% |
| **+ Full Combination** | **26 (13.0%)** | **20 (10.0%)** | **151 (75.5%)** | **-24.5%** |

**Cost-Benefit Analysis.** While the mitigation strategies significantly improve robustness, they introduce marginal computational overhead. As shown in Table 13, the "Full" configuration increases inference latency by 7.5%, which is a favorable trade-off for the 24.5% reduction in failure rate.

Table 13: Computational cost analysis of mitigation strategies.

| Configuration | Latency (ms/token) | Rel. Increase | Memory (GB) |
|---|---|---|---|
| Baseline | 14.7 | - | 8.4 |
| Full Improved | 15.8 | +7.5% | 8.6 |

**Discussion.** The results demonstrate that the majority of architectural failures are addressable through inference-time optimization without retraining. The remaining failures (75.5% of original set) are primarily driven by Long Context Degradation (12.5%) and Cultural Context Misinterpretation (9.0%), which require data-centric solutions rather than architectural changes.

## H  GENERALIZATION AND EXTENSIBILITY ANALYSIS

A fundamental question for specialized architectures is their ability to generalize beyond training distributions. We evaluate KSKT on two challenging out-of-distribution tasks: linguistic style imitation (verifying generalization beyond personality traits) and emergent complex role modeling (verifying SAMOE's extensibility).

### H.1  LINGUISTIC STYLE IMITATION

To test whether KSKT's dual-perspective reasoning extends to linguistic constraints, we conducted a proof-of-concept experiment by fine-tuning the model on a Shakespearean dialogue dataset (5.6% of total training data). The goal is to balance *Archaic Style Consistency* (Self-Stream constraint) with *Modern User Responsiveness* (Other-Stream requirement).

**Results.** Table 14 compares KSKT against the base model and GPT-4 (prompted).

Table 14: Performance on Shakespearean Style Imitation. KSKT achieves near-perfect balance ($|\Delta| = 0.07$) between style and responsiveness, significantly outperforming the base model which sacrifices style for clarity. Training cost for KSKT is stable compared to inference-heavy prompting.

| Model | Style Consistency | User Responsiveness | Balance ($|\Delta|$) | Protocol |
|---|---|---|---|---|
| Qwen3-Base | $3.52 \pm 0.12$ | $3.94 \pm 0.10$ | 0.42 | Fine-tuning |
| GPT-4 (0-shot) | $3.85 \pm 0.11$ | $4.21 \pm 0.08$ | 0.36 | Prompting |
| GPT-4 (1-shot) | $4.02 \pm 0.09$ | **$4.28 \pm 0.07$** | 0.26 | In-context |
| **KSKT (Ours)** | **$4.15 \pm 0.08$** | $4.08 \pm 0.09$ | **0.07** | Fine-tuning |

**Analysis.** KSKT demonstrates a +15.1% relative gain in Style Consistency over the base model (4.15 vs. 3.52). Crucially, while the base model acts conservatively by reverting to modern English (Balance=0.42), KSKT maintains strong dual-perspective alignment, validating that the architecture treats "Style" as a constraint analogous to "Personality."

### H.2  SAMOE EXTENSIBILITY VIA TOP-$k$ ROUTING

We investigate whether the fixed expert set $(P, K, E, C)$ limits the modeling of complex, multi-dimensional characters (e.g., an *Empathetic Stoic* or *Ironic Narrator*). We introduce a **Top-$k$ Weighted Routing** strategy ($k = 2$) to enable combinatorial expert usage.

**Design.** We constructed a "Complex Role Set" comprising 15% of test cases, featuring characters with conflicting internal traits (e.g., a *Utilitarian Doctor* needing both empathy and cold logic). We compare the default Top-1 routing against a Top-2 weighted ensemble.

**Expert Synergy Analysis.** As illustrated in Figure 14, expert activation patterns shift from single-expert dominance to collaborative distributions for complex roles. For instance, in "Empathetic

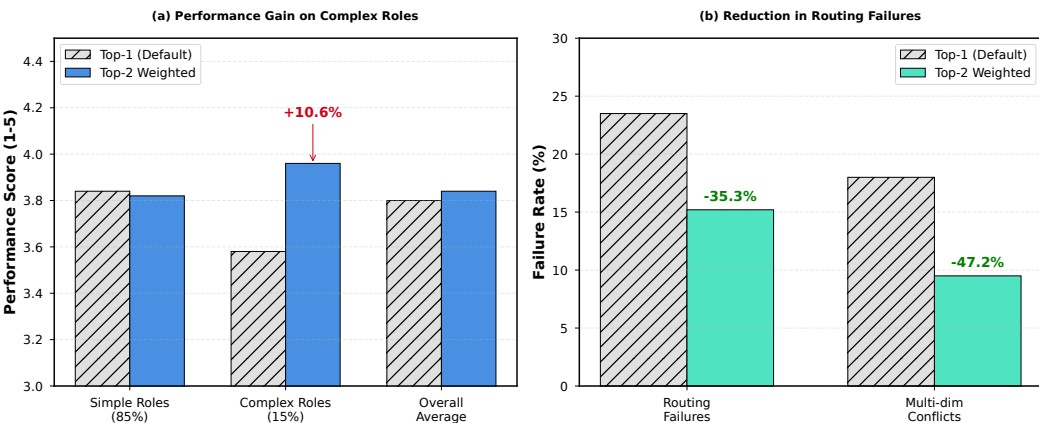

Figure 14: Impact of Top-2 Routing on complex roles. **(a)** While Top-2 routing introduces slight noise for simple roles (-0.5%), it unlocks a massive +10.6% gain for complex roles by allowing experts to collaborate (e.g., Personality + Emotion). **(b)** This collaborative mechanism significantly reduces routing failures (-35.3%) and multi-dimensional conflicts (-47.2%), validating the architecture's extensibility.

Stoic" scenarios, the activation changes from $E : 72\%$ (Top-1, forcing a choice between empathy and philosophy) to a balanced $E : 52\% + P : 38\%$ (Top-2). This confirms that SAMOE supports **emergent role modeling** through combinatorial generalization (Fedus et al., 2022), refuting concerns about expert rigidity. Based on these findings, we recommend Top-1 for general efficiency and Top-2 for handling complex, multi-faceted personas.

## I   REPRODUCIBILITY STATEMENT

To ensure full reproducibility of our results, we provide the following:

**Code and Data Availability.** Complete implementation code will be released upon acceptance, including all architectural components, training scripts, and evaluation protocols. Training data construction pipeline and character profiles will be made available following ethical guidelines.

**Experimental Configuration.** All hyperparameters are documented in Table 3. Hardware specifications: 8×NVIDIA V100-32GB GPUs, PyTorch 1.13.0, transformers 4.21.0, CUDA 11.7.

**Random Seed Configuration.** Fixed seeds ensure reproducible results: training (42), evaluation (2023), data sampling (12345). All experiments use deterministic operations where possible.

**Evaluation Protocols.** Detailed evaluation scripts with exact metrics computation, statistical testing procedures, and human evaluation guidelines will be provided.

**Model Checkpoints.** Final trained model checkpoints will be made available through appropriate channels following publication, enabling direct result reproduction and comparison studies.

## LARGE LANGUAGE MODEL USAGE STATEMENT

In the interest of transparency and in compliance with ICLR 2026's LLM usage policies, we provide a detailed account of how large language models assisted in this research. We emphasize that all technical contributions, experimental designs, and scientific insights presented in this work are entirely our own, and we take full responsibility for all content.

We employed several state-of-the-art language models to enhance different aspects of our research process while maintaining the integrity of our core contributions. Specifically, we utilized Claude Sonnet 4's advanced research capabilities to conduct comprehensive literature reviews and identify relevant prior work in role-playing language models and dual-perspective reasoning. This assisted us in ensuring comprehensive coverage of the existing literature and identifying key research gaps,

though all synthesis, analysis, and positioning of our work relative to prior art remained our original contribution.

For manuscript preparation, we employed GPT-5 to improve the grammatical accuracy, sentence structure, and overall clarity of our writing. The model helped optimize word choices and enhance the flow of technical explanations while preserving the original meaning and technical accuracy of our content. All mathematical formulations, experimental procedures, and theoretical frameworks were developed independently by the authors.

Additionally, we used Google's Gemini 2.5 Flash Image (code-named "nano-banana") to assist in conceptualizing and refining the visual representation of our methodology. The model helped us explore different ways to present our architectural components clearly and effectively, contributing to the final methodology diagram design. However, all architectural decisions and technical specifications depicted in the figures represent our original research contributions.

We believe that transparent disclosure of AI assistance, as mandated by ICLR 2026, strengthens the scientific process and sets a positive precedent for the responsible integration of AI tools in academic research. The use of these tools enhanced the clarity and accessibility of our work without compromising the originality or integrity of our scientific contributions.

