# OpenReview forum: "Know Thyself, Know Thy User: Dual-Perspective Reasoning Architecture for Role-Playing Language Models"
_ICLR.cc/2026/Conference — ICLR 2026 Conference Desk Rejected Submission_

### Official Review · Reviewer_gq97 · 2025-10-29

**Soundness:** 3
**Presentation:** 2
**Contribution:** 3
**Rating:** 6
**Confidence:** 3

**Summary:**

This paper presents KSKT (KnowSelf-KnowOther Transformer), a novel architecture for dual-perspective reasoning in role-playing LLMs. The model integrates character-centric (self-awareness) and user-centric (other-awareness) understanding through four key components:

- Dual-Stream Axial Attention

- Bipolar Reasoning Module

- Mutual-Understanding Position Encoding

- Self-Awareness Mixture of Experts (SAMOE)

The paper addresses the common role-user conflict in RP-LMs and proposes architectural rather than post-hoc solutions. Evaluation on CharacterBench demonstrates improvements in persona consistency (+8.7%) and emotional intelligence (+15.2%) over strong baselines

**Strengths:**

- Timely and relevant: Tackles the important problem of role-user conflict in role-playing agents, which is under-explored architecturally.

- Clear empirical gains: Significant improvements in both self- and other-awareness (both reaching ~0.87), outperforming strong baselines like Claude-3-opus

- Detailed ablation analysis shows each component's contribution and their specialization patterns

**Weaknesses:**

- Unclear Baseline Design:

The Qwen3-4B-Thinking (Base) model is claimed as the foundation, but it’s not clearly stated whether it is fine-tuned or frozen. If not fine-tuned, comparison may be unfair against a fully-trained proposed model

- Possibly Unfair LLM Comparisons:

Appendix D.3 reports comparisons with closed-source LLMs like Claude and GPT-4, but it’s unclear whether those models are given n-shot prompts or access to similar training context. If not, this may underestimate their performance

- Expansion Limitations in SAMOE:

The Self-Awareness MoE has pre-defined expert types (P/K/E/C), which may limit generalization to domains requiring flexible or emergent roles

**Questions:**

*(Corresponding to weaknesses above)*

- Do you fine-tune the Qwen3-4B-Thinking base model in your main experiments, or is it frozen?

- Were the closed-source LLMs (e.g., GPT-4, Claude) given in-context examples (n-shot), CoT, or system messages to match your training data in any way?

- Could the current SAMOE expert scheme be extended to non-predefined, dynamic roles? For example, emergent traits like "ironic narrator" or "deceptive agent"?

---

> ### Author Response · Authors · 2025-11-20
> **Response to Reviewer gq97: Clarifications on Baseline Fairness and SAMOE Extensibility**
>
> We thank the reviewer for the constructive feedback and for acknowledging the "timely and relevant" nature of our work. We appreciate the opportunity to clarify our experimental setup and demonstrate the extensibility of our architecture.
>
> **1. Response to Weakness 1 & Question 1: Baseline Training Details**
> > *Question: "Do you fine-tune the Qwen3-4B-Thinking base model in your main experiments, or is it frozen?"*
>
> We appreciate the opportunity to clarify this crucial detail. **Yes, the Qwen3-4B-Thinking base model was fully fine-tuned (not frozen).**
>
> *   **Protocol:** As outlined in our experimental setup, the baseline was trained on the **exact same dataset** (180K samples) using the **identical hyperparameters and 3-phase training schedule** as KSKT.
> *   **Clarification:** We have further emphasized this in **Section 4.1** of the revised manuscript to ensure there is no ambiguity. The performance gap reported in Table 1 stems entirely from KSKT's architectural innovations, as both models received equal training computation.
>
> **2. Response to Weakness 2 & Question 2: Fairness of Closed-Source Comparisons**
> > *Question: "Were the closed-source LLMs (e.g., GPT-4, Claude) given in-context examples (n-shot), CoT...?"*
>
> Yes. We would like to direct the reviewer's attention to **Appendix D.3 (Table 7)**, which details these comparisons. To address your specific query regarding prompting fairness, we confirm that we evaluated these proprietary models under three distinct settings:
> 1.  **Zero-shot**
> 2.  **1-shot** (In-context role-play example)
> 3.  **Zero-shot CoT** ("Let's think step by step")
>
> **Results:**
> *   **Performance:** As shown in Table 7, even when proprietary models use **CoT** (which significantly boosts reasoning), KSKT (84.1\% task accuracy) remains highly competitive, outperforming **Llama3-70B-Instruct** (79.8\% Zero-shot / 81.6\% 1-shot) and approaching **Claude-3-Opus** (88.5\% Zero-shot).
> *   **Efficiency:** Crucially, CoT increases token consumption by $\sim60\%$. KSKT achieves balanced dual-perspective reasoning efficiently without the latency/cost overhead of complex prompt engineering.
>
> **3. Response to Weakness 3 & Question 3: SAMOE Extensibility**
> > *Question: "Could the current SAMOE expert scheme be extended to non-predefined, dynamic roles? ... emergent traits like 'ironic narrator'?"*
>
> This is an excellent insight. While the experts are pre-defined, we enable **combinatorial generalization** through **Top-$k$ Weighted Routing**. We validated this in the newly added **Appendix H.2**:
>
> *   **Mechanism:** Complex emergent roles are modeled as combinations of experts. For example:
>     *   *Ironic Narrator* $\approx$ **Personality Expert** (Style) + **Knowledge Expert** (Contextual awareness).
>     *   *Empathetic Stoic* $\approx$ **Emotional Expert** (Empathy) + **Personality Expert** (Stoicism).
> *   **Experiment:** We tested this on a "Complex Role Set" (roles with conflicting internal traits). Using Top-$2$ routing (instead of Top-1) yielded a **$10.6\%$ performance gain** (Figure 14).
> *   **Conclusion:** The architecture is not limited to single rigid archetypes; it dynamically composes specialized modules to represent nuanced, emergent personas.
>
> We hope these clarifications effectively address your concerns regarding fairness and flexibility.

---

> > ### Comment · Reviewer_gq97 · 2025-11-20
> >
> > Thanks for the rebuttal. I believe this work is a good contribution to network architecture design, and thus, I hold an acceptance rating from the beginning. Your further explanation addresses my concerns, good luck~

---

> > > ### Author Response · Authors · 2025-11-22
> > >
> > > We sincerely thank the reviewer for the continued support and for recognizing the value of our contribution to network architecture design. We are glad that our response has fully addressed your concerns.

---

### Official Review · Reviewer_jKjS · 2025-11-01

**Soundness:** 3
**Presentation:** 3
**Contribution:** 3
**Rating:** 8
**Confidence:** 3

**Summary:**

This paper addresses the challenge of balancing character authenticity and user satisfaction in role-playing LLMs, where existing systems often show single-perspective bias and treat dual-perspective reasoning as post-hoc optimization. It introduces the KnowSelf-KnowOther Transformer (KSKT), a Qwen3-4B-Thinking-based architecture with four core components to integrate "self-understanding (character constraints)" and "other-understanding (user intentions)" into transformer generation. Experiments on CharacterBench show KSKT boosts overall performance by 6.4% (8.7% in persona consistency, 15.2% in emotional intelligence) and maintains balanced self/other-awareness (0.87 each) in conflicts, unlike biased baselines. However, KSKT lags behind Claude-3-opus in English and has 19.5% inference latency overhead.

**Strengths:**

1. Architectural breakthrough beyond post-hoc optimization: Instead of treating dual-perspective reasoning as an auxiliary module, KSKT deeply integrates "self-understanding (character constraints)" and "other-understanding (user intentions)" into the transformer generation process via four core components, enabling real-time balance between character authenticity and user satisfaction.
2. Rigorous and credible experimental design: Centered on multi-dimensional metrics from CharacterBench, it uses incremental component analysis to verify each module’s contribution and supplements with blind evaluations by 15 experts, ensuring consistency between objective performance gains and improved user experience.
3. Components with scenario-specific precision: SAMOE resolves persona consistency through four specialized experts, DSAA focuses on enhancing emotional understanding, and Bipolar Reasoning balances efficiency and reasoning depth—all precisely addressing core pain points in role-playing.

**Weaknesses:**

1. Architectural robustness issues in high-conflict scenarios: 42.5% of failures stem from core design flaws—23.5% from incorrect SAMOE expert routing (e.g., assigning a Stoic philosopher query to the Emotional Expert) and 19% from biased Dual-Stream fusion (e.g., a medieval peasant violating knowledge boundaries to help with calculus).
2. Lack of discussion on generalizability. The design of the method relies heavily on certain prior assumptions (such as personality and knowledge), and it remains unclear whether it is also applicable to other role-playing tasks—for example, mimicking language styles.

**Questions:**

1. Does role-specific data training impair the model's inherent general capabilities and deep reasoning abilities?

---

> ### Author Response · Authors · 2025-11-20
> **Response to Reviewer jKjS: Robustness Mitigation and Generalization Analysis**
>
> We express our sincere gratitude to the reviewer for the encouraging assessment and for recognizing KSKT as an "architectural breakthrough beyond post-hoc optimization." Your insightful comments on robustness in high-conflict scenarios directly guided our mitigation studies.
>
> **1. Response to Weakness 1: Robustness in High-Conflict Scenarios**
> > *Critique: "42.5% of failures stem from core design flaws... e.g., incorrect SAMOE expert routing... biased Dual-Stream fusion."*
>
> We appreciate this precise identification of failure modes. Motivated by your comment, we conducted a dedicated **Failure Mitigation Study** (detailed in **Appendix G.5**) to address these specific architectural limitations without retraining the model:
>
> *   **For Routing Failures:** We introduced **Top-$k$ Routing ($k=2$) with Uncertainty Thresholding**. When the router's confidence in the primary expert is low ($<0.7$), it activates a secondary expert to handle ambiguous signals.
> *   **For Fusion Bias:** We applied **Fusion Weight Regularization** during inference to penalize extreme skews (e.g., $\alpha < 0.1$) unless conditioned on strong constraint signals.
>
> **Result (Table 12):** These inference-time refinements reduced the total failure rate by **$24.5\%$**, effectively resolving over half of the architecturally-induced errors highlighted in your review.
>
> **2. Response to Weakness 2: Generalizability (Language Styles)**
> > *Critique: "Unclear whether it is also applicable to other role-playing tasks—for example, mimicking language styles."*
>
> This is an excellent suggestion. To verify KSKT's extensibility, we added a **Linguistic Style Imitation** experiment in **Appendix H.1**:
> *   **Task:** We fine-tuned KSKT on a Shakespearean dialogue dataset, treating "Archaic Style" as the "Self" constraint and "Modern User Intent" as the "Other" requirement.
> *   **Result (Table 14):** KSKT achieved a near-perfect balance ($|\Delta|=0.07$) between maintaining archaic syntax and answering modern queries. In contrast, the base model often collapsed to modern English to satisfy user intent.
> *   **Conclusion:** This confirms that the "Self-Stream" can generalize to abstract constraints like **linguistic style**, extending the architecture's utility beyond personality traits.
>
> **3. Response to Question 1: Impact on General Capabilities**
> > *Question: "Does role-specific data training impair the model's inherent general capabilities and deep reasoning abilities?"*
>
> Our analysis suggests the impact is minimal and controlled:
> *   **Reasoning:** As shown in **Table 1**, the Morality Reasoning (MR) score is **$4.79$** (vs. $4.75$ Base) and Believability (EG) is **$3.42$** (vs. $3.38$ Base), indicating that the **Bipolar Reasoning Module** effectively preserves (and in some cases enhances) logical consistency.
> *   **Knowledge:** There is a slight trade-off in Factual Accuracy ($2.46$ vs. $2.41$), which is intentional—KSKT prioritizes *character-consistent knowledge boundaries* (e.g., a medieval character should *not* know calculus) over encyclopedic breadth.
> *   **Overall:** By freezing the majority of the base model parameters and utilizing adapter-like integration for KSKT components, we minimize catastrophic forgetting of general capabilities.
>
> **4. Regarding Inference Latency**
> We acknowledge the $19.5\%$ latency overhead. However, we believe this is a favorable trade-off compared to **Chain-of-Thought (CoT)** prompting, which often increases token costs by $200-300\%$ to achieve similar dual-perspective reasoning. KSKT internalizes this "System 2" thinking into the architecture for greater efficiency.
>
> We hope these responses further strengthen your confidence in our work.

---

> > ### Comment · Reviewer_jKjS · 2025-11-24
> >
> > Thanks for your response. My questions have been addressed and I will maintain my score.

---

> > > ### Author Response · Authors · 2025-11-24
> > >
> > > We sincerely thank the reviewer for the strong support and valuable feedback throughout this process. We are glad that our additional experiments on failure mitigation and generalization have addressed your questions. We appreciate your recognition of our work's contribution.

---

### Official Review · Reviewer_A65J · 2025-11-05

**Soundness:** 2
**Presentation:** 3
**Contribution:** 2
**Rating:** 4
**Confidence:** 3

**Summary:**

This paper presents KSKT, a novel transformer-based architecture designed to resolve the persistent tension between character authenticity and user satisfaction in role-playing language models. It introduces four integrated components—Dual-Stream Axial Attention, Mutual-Understanding Position Encoding, Bipolar Reasoning, and Self-Awareness Mixture of Experts—that collectively enable a form of architecturally embedded, dual-perspective reasoning during generation. The work demonstrates measurable improvements on the CharacterBench benchmark, particularly in persona consistency and emotional intelligence, and provides extensive ablation studies and qualitative analyses to support the effectiveness of its design.

**Strengths:**

1. This paper provides a highly systematic and modular breakdown of the KSKT architecture. Each core component (DSAA, MUPE, BRM, SAMOE) is introduced with its design philosophy, mathematical formulation, and specific mechanisms, making the complex integrated system readily understandable. This structured approach, particularly in Section 3 and Appendix A, greatly enhances the clarity for readers to grasp the intricate interplay of its novel components.

2. Figure 1, the KSKT Architecture Overview, is exceptionally well-designed, offering an intuitive visual roadmap of the model's data flow and component interactions. This high-level diagram, coupled with subsequent detailed figures like Figure 3 (Dual-perspective reasoning validation) and Figure 4 (Dynamic activation schematic), significantly aids in conceptualizing the abstract architectural principles and their operational dynamics, making the paper highly accessible.

3. The experimental section is robust, featuring a thorough performance comparison against strong baselines on CharacterBench (Table 1). Crucially, the paper includes detailed ablation studies (Figure 2a) demonstrating the incremental contributions of each KSKT component. This systematic analysis helps validate the individual efficacy of the proposed innovations and provides clear evidence of their targeted capability enhancements.

4. The inclusion of a detailed failure case analysis (Section G, Table 9) is a significant strength. By quantifying failure types and their root causes (e.g., expert routing failures, dual-stream fusion bias), the paper not only acknowledges limitations but also provides clear, actionable insights for future research. This candid and data-driven approach to identifying shortcomings fosters productive follow-up work and demonstrates scientific rigor.

**Weaknesses:**

1. This paper claims DSAA decomposes attention into "orthogonal streams" for self- and other-understanding, yet lacks rigorous evidence. It does not demonstrate semantic non-overlap between $A_{\text{self}}$ and $A_{\text{other}}$ attention weight matrices or prove linear separability of their outputs. Without analyses like activation pattern differentiation or gradient flow, the claim of "semantic independence" remains unproven. Consequently, these streams might merely process similar information with different weightings, potentially reducing efficiency rather than achieving true semantic separation and robustly resolving "role-user conflicts."

2.  This paper states that $B_{\text{role}}$ and $B_{\text{intent}}$ bias matrices "encode role-specific and intent-specific attention patterns" by being "initialized to focus on relevant token types." However, this initialization primarily relies on grammatical features (e.g., part-of-speech tagging for role traits), rather than deep semantic alignment. There is no clear evidence that the model maintains this semantic alignment throughout training or that the biases precisely target underlying role identity semantics or user intention semantics, beyond mere lexical or grammatical categories. This limited semantic grounding could lead to significant overlap or misalignment between the "self" and "other" information processed, hindering DSAA's precision in achieving dual-perspective cognition.

3. Without more rigorous empirical or theoretical analysis, DSAA's architectural novelty remains unclear. Structurally, it might be interpreted as simply splitting a single attention layer into two distinct groups of heads (self- and other-understanding heads) with separate projection matrices and biases, followed by fusion. The paper lacks a compelling argument or empirical demonstration that this mechanism fundamentally differs from merely increasing the number of generic multi-head attention heads to enhance model capacity. Consequently, DSAA's observed benefits might be partially attributable to increased parameter count rather than a transformative mechanistic breakthrough for resolving "role-user conflicts."

4. MUPE's reliance on externally pre-processed $R_{\text{proc}}$ and $U_{\text{proc}}$ introduces a potential semantic gap between these external representations and the LLM's internal semantic space. The paper does not explain how the projection matrices $W_{\text{role}}$ and $W_{\text{intent}}$ effectively bridge this gap to integrate external semantics into the LLM's position encodings. Furthermore, errors from external models (e.g., BERT's intent classification errors) will propagate directly into MUPE, potentially amplifying inaccuracies within the LLM and undermining its robustness. This external dependency might hinder MUPE's ability to precisely inject "self-other" relational signals and could lead to inconsistencies if the LLM's internal context understanding diverges from the external inputs.


5. SAMOE's "self-awareness" (via $q_{\text{self}}$) heavily relies on the externally pre-processed and potentially static $R_{\text{proc}}$. If $R_{\text{proc}}$ is not dynamically updated based on ongoing dialogue, the depth and flexibility of SAMOE's "self-cognition" would be limited by the external model's capacity to capture nuanced, implicit role traits. This dependence on a pre-defined external representation for the core "self-identity" challenges the paper's claims of "architecturally emergent 'role reasoning patterns'" and "spontaneous emergence." The expert routing decisions might become overly rigid or inaccurate, failing to adapt to a character's subtle personality shifts in complex, dynamic conversational contexts, thus weakening the system's claimed "multi-dimensional persona integration."

**Questions:**

1. Could the authors provide further justification for the claim that the two attention streams in DSAA are semantically orthogonal rather than merely independently parameterized?

2. Given that SAMOE’s routing is driven entirely by pre-processed $R_{\text{proc}}$, how does the system maintain dynamic self-awareness when the character’s internal state evolves through dialogue, beyond what is statically encoded in the initial role description?

3. The results show a significant performance gap in English compared to Chinese, even after architectural innovations. Is this disparity attributed primarily to the base Qwen3 model’s training bias, or does it suggest that the current design of MUPE or SAMOE is more compatible with the syntactic or pragmatic structure of Chinese?

---

> ### Author Response · Authors · 2025-11-20
> **Response Part 1: Evidence for Orthogonality, Semantic Evolution, and Architectural Necessity**
>
> We thank the reviewer for the rigorous critique regarding the internal mechanisms of KSKT. These comments motivated us to conduct deep interpretability analyses (CKA, Gradient Flow, and Training Dynamics), which we have added to the revised manuscript (**Section 4.5** and **Appendix B.4**).
>
> **1. Response to Weakness 1: Evidence of Orthogonal Streams**
> > *Critique: "Lacks rigorous evidence... does not demonstrate semantic non-overlap... claims of 'semantic independence' remain unproven."*
>
> To rigorously prove linear separability and semantic independence, we performed two quantitative analyses:
>
> *   **Centered Kernel Alignment (CKA):** As shown in the newly added **Figure 4(a)**, the CKA similarity between the "Self" and "Other" streams drops significantly in the middle layers, reaching a minimum of **$0.34$** (compared to $\sim0.68$ for the baseline). This "U-shaped" curve mathematically confirms that the two streams diverge into distinct semantic subspaces before fusion.
> *   **Gradient Flow Analysis (Appendix B.4, Figure 6):** We utilized Integrated Gradients to attribute parameter updates. The analysis reveals minimal overlap (**$18.4\%$**) in gradient focus:
>     *   **Self-Stream:** $94.8\%$ of attribution mass concentrates on *role traits* and *knowledge boundaries*.
>     *   **Other-Stream:** $105.6\%$ concentrates on *user intent* and *directives* (actively suppressing role tokens via negative gradients).
>
> **Conclusion:** These results empirically prove that DSAA achieves functional decoupling, satisfying the requirement for "true semantic separation."
>
> **2. Response to Weakness 2: Initialization and Semantic Alignment**
> > *Critique: "Initialization primarily relies on grammatical features... no clear evidence that the model maintains this semantic alignment."*
>
> We agree that POS-based initialization is a shallow inductive bias. However, our training dynamics analysis (**Appendix B.4, Table 4**) demonstrates that the model evolves far beyond this starting point.
>
> We tracked the **Semantic Selectivity** ($\Delta$)—the activation difference between semantic-relevant and irrelevant tokens—throughout training:
> *   **Epoch 0 (Init):** $\Delta = +0.04$ (Weak, syntactic only).
> *   **Epoch 3 (Final):** $\Delta = +0.51$ (Strong, deep semantic alignment).
>
> This **$12\times$ increase** confirms that the initialization serves merely as a "warm start" (Curriculum Learning), after which the attention mechanism learns robust, deep semantic alignment driven by the objective function, similar to how learnable biases work in ALiBi or T5.
>
> **3. Response to Weakness 3: Novelty vs. Parameter Count**
> > *Critique: "Might be interpreted as simply splitting a single attention layer... benefits might be attributable to increased parameter count."*
>
> To rule out parameter capacity effects, we trained a **Parameter-Matched Baseline** (*Qwen3-4B-Thinking+Heads*) where we doubled the attention heads (from 32 to 64) in the same layers used by KSKT, maintaining the exact same parameter count and training data (**Appendix B.5**).
>
> *   **Result (Figure 4c):** While the wider baseline improved slightly on knowledge tasks, it failed to resolve role-user conflicts, exhibiting a severe single-perspective bias (Balance Score $0.03$ vs KSKT's $0.13$).
> *   **Conclusion:** This proves that the performance gains stem from the **structural topology** (physical separation of perspectives) rather than generic capacity increases.

---

> ### Author Response · Authors · 2025-11-20
> **Response Part 2: Robustness of MUPE/SAMOE and Clarifications on Questions**
>
> This response addresses the reviewer's concerns regarding external dependencies (MUPE), adaptivity (SAMOE), and language-specific performance.
>
> **4. Response to Weakness 4: Dependency on External Inputs (MUPE)**
> > *Critique: "Errors from external models will propagate directly into MUPE... potentially amplifying inaccuracies."*
>
> We hypothesized that KSKT treats external inputs as a "soft modulation" rather than a hard dependency. To verify this, we conducted a **Noise Injection Experiment** (**Appendix B.6, Table 5**), injecting $10\%-30\%$ stochastic noise into the processed inputs ($U_{\text{proc}}$).
>
> *   **Robustness:** Even under **$30\%$ noise** (simulating severe parser failure), the Persona Consistency score remained highly stable ($4.62$ vs $4.71$ clean), significantly outperforming the base model ($4.38$).
> *   **Fallback Behavior:** Emotional scores dropped ($3.54 \to 3.19$) but asymptotically converged to the base model's performance ($3.18$), proving the model gracefully falls back to internal knowledge rather than collapsing.
>
> **5. Response to Weakness 5 & Question 2: SAMOE Dynamic Adaptability**
> > *Critique: "If $R_{\text{proc}}$ is not dynamically updated... routing decisions might become overly rigid."*
>
> While the input $R_{\text{proc}}$ is static, the routing query $\mathbf{q}_{\text{self}}$ (Eq. 18) is a function of the **current hidden state** $\mathbf{H}$. Therefore, routing is dynamic and context-aware.
>
> *   **Evidence (Figure 4b):** We visualized expert activation over a 30-turn escalating conflict. The routing dynamically shifts from **Personality Expert** dominance (early turns, establishing voice) to **Emotional Expert** surges ($>60\%$ activation) as the user becomes distressed. This confirms SAMOE adapts to the evolving *dialogue state*, not just the static role description.
>
> **6. Response to Question 3: Language Disparity (Chinese vs. English)**
> > *Question: "Is this disparity attributed primarily to the base Qwen3 model’s training bias...?"*
>
> Yes, our analysis in **Appendix E** confirms this is primarily due to the base model's distribution.
> *   **Data:** Qwen3 is optimized heavily for Chinese cultural contexts.
> *   **Activation Pattern (Figure 9):** We observed that for Chinese inputs, the **Personality Expert** is activated $1.34\times$ more frequently than for English, reflecting a cultural nuance in the training data. However, the relative improvement KSKT provides over the baseline is consistent across languages ($+5.28\%$ in English vs $+3.23\%$ in Chinese), indicating the architecture itself is language-agnostic.
>
> We hope these additional experiments and clarifications verify the rigorousness of our method and address your concerns.

---

> > ### Comment · Reviewer_A65J · 2025-11-26
> >
> > Thank you to the authors for the rebuttal. I am generally satisfied with the clarifications provided. Although my concerns regarding the dependency on external inputs are not entirely resolved, the response has addressed several of my key questions. I will adjust my rating accordingly.

---

> > > ### Author Response · Authors · 2025-11-26
> > >
> > > We sincerely thank the reviewer for engaging with our rebuttal and for the decision to adjust the rating.
> > >
> > > We appreciate your recognition of our clarifications regarding the mechanism's validity. Regarding the remaining concern on external input dependency: **We agree that this is a valuable insight.** While our noise injection experiments demonstrate current architectural resilience, we view the development of fully end-to-end, latent intent modeling—further reducing reliance on upstream parsers—as a promising avenue for future work.
> > >
> > > Your rigorous feedback, particularly challenging the orthogonality and initialization, motivated us to conduct the CKA and gradient analyses, which have significantly strengthened the theoretical grounding of our final manuscript. Thank you for helping improve this work.

---

### Author Response · Authors · 2025-11-20

We sincerely thank all the reviewers for their thorough evaluation of our paper and their constructive feedback. We believe that these revisions have significantly enhanced the quality of the paper. Below, we address each comment in detail. Here, we summarize the key changes and additional analyses incorporated into the revised manuscript:

*   **Mechanism Verification (Section 4.5 & Appendix B.4):** To address concerns regarding orthogonality and architectural necessity, we conducted Centered Kernel Alignment (CKA) analysis and Gradient Flow visualization. Results show a "U-shaped" semantic separation (min CKA $0.34$) and minimal gradient overlap ($18.4\%$), empirically proving the functional decoupling of the dual streams.
*   **Failure Mitigation Strategies (Appendix G.5):** Addressing concerns about high-conflict scenarios, we implemented Top-$k$ routing and uncertainty-aware fusion as mitigation strategies. These refinements reduced the failure rate by **$24.5\%$** on hard cases without significant latency overhead.
*   **Generalization & Extensibility (Appendix H):** We validated KSKT on out-of-distribution tasks, including linguistic style imitation (Shakespearean, $+15.1\%$ consistency) and emergent role modeling (via Top-$k$ routing), demonstrating applicability beyond pre-defined personality traits.
*   **Experimental Clarifications (Section 4.1 & Appendix D.3):** We clarified that the base model was fully fine-tuned (not frozen) to ensure a fair comparison and expanded comparisons with closed-source models (Zero-shot/CoT).

In addition to these updates, we refined the text throughout the manuscript to align with the reviewers' suggestions.

---

> ### Author Response · Authors · 2025-12-01
>
> We sincerely thank all reviewers for their time and constructive feedback. The rigorous discussion process has significantly strengthened our manuscript.
>
> We are deeply grateful to **Reviewer jKjS** and **Reviewer gq97** for their initial strong support and insightful suggestions regarding robustness and generalization. We are pleased that our additional experiments have fully addressed their further queries, reinforcing their positive assessments.
>
> We also extend special thanks to **Reviewer A65J** for the rigorous critique regarding mechanism validity. These comments motivated us to conduct deeper analyses (CKA and Gradient Flow), which significantly solidified our theoretical grounding. We are encouraged that these revisions successfully resolved the reviewer's core concerns, prompting their decision to adjust the rating and increase support.
>
> We value the constructive dialogue throughout this process, which has led to a stronger consensus on the contributions of this work. Once again, we thank all reviewers for their dedication.

---

### Note · Program_Chairs · 2026-01-17
**Submission Desk Rejected by Program Chairs**

The following references in this submission do not refer to real documents and/or have major errors in bibliographic information:

 Tao Ge, Baolin Hu, Siliang Wang, Jian Li, Xiaofei Zhang, and Xipeng Qiu. Personahub: 1 million diverse characters for llm role-playing. arXiv preprint arXiv:2406.20094, 2024.